# METRIS: MULTI-EXPRESSIONS FOR TRANSFORMER-BASED REFERRING IMAGE SEGMENTATION

## ABSTRACT

Referring image segmentation (RIS) aims to precisely segment a target object described by a linguistic expression. Recent RIS methods have introduced Transformer-based networks that use vision features as query and linguistic expression features as key-value to find target regions by referring to the given linguistic information. Since the Transformer-based network predicts based on the guidance information that guides the network on which regions to pay attention, the capacity of this guidance information has a significant impact on segmentation results in Transformer-based RIS. However, existing methods rely only on linguistic tokens as the guidance elements, which are limited in providing the visual understanding of the fine-grained target regions. To address this issue, we present a novel **M**ulti-**E**xpression guidance framework for **T**ransformer-based **R**eferring **I**mage **S**egmentation, METRIS, which allows the network to refer to the *visual expression* tokens as the guidance information alongside the linguistic expression tokens. The introduction of visual expression can complement the capability of linguistic guidance by effectively providing the target-informative visual contexts. To generate the visual expression from vision features, we introduce a visual expression extractor that is designed to endow with the *target-oriented visual guidance ability* and to acquire rich contextual information. This module strengthens the adaptability to the diverse image and language inputs, and improves visual understanding of the fine-grained target regions. Extensive experiments demonstrate the effectiveness of our approach across the commonly used RIS settings and the generalizability evaluation settings. Our method consistently shows strong performance on three public RIS benchmarks.

## 1 INTRODUCTION

Referring image segmentation (RIS) (Hu et al., 2016; Chen et al., 2022) is one of the challenging vision-language tasks (Yan et al., 2023; Ghosh et al., 2024; Chen et al., 2024b; Hu et al., 2024), and can be applied in various applications such as human-robot interaction and the object retrieval. Given an image and a natural language expression describing a target object within the image, one of the key points in this task is for the network to precisely segment the target object regions from the image by referring to the given expression. With the great success of Transformer-based networks (Vaswani, 2017; Dosovitskiy et al., 2020) in single modal segmentation tasks (Qian et al., 2023; Zhou & Wang, 2024; Liu et al., 2024b), Transformer-based methods have been actively studied on RIS task. To find specific regions by referring to the given information, RIS models use vision features as query and the given information as key-value in the Transformer network, as shown in Figure 1; the set of such information provided to the Transformer network as key-value is called *Guidance Set* in this paper. Specifically, the role of the guidance set is to guide the network on which regions to focus its attention, and the network predicts target regions based on the guidance information. Motivated by this fact, we focus on that enhancing the capability of the guidance set has a significant impact on segmentation performance in Transformer-based referring image segmentation.

Most previous works have approached this task by directly enhancing the language features to improve the comprehension for the language expression. Some of these studies (Ding et al., 2022a; Hu et al., 2023) obtain the enhanced linguistic features by allowing language features to refer to vision features via the language-vision cross-attention mechanism (Figure 1 (b)). More recent studies (Lai

Figure 1: Illustration of different guidance sets. Unlike previous approaches, our approach allows visual expression, which is equipped with target-informative visual guidance ability, to be used as guidance elements to enhance the guidance capability for Transformer-based referring image segmentation.

et al., 2024; Ren et al., 2024) employ large language models (LLMs) (Touvron et al., 2023; Chiang et al., 2023; Liu et al., 2024a) to improve the understanding of the language expression via LLM's immense knowledge, and exploit the generated language token in the segmentation network (Figure 1 (c)). These existing studies successfully have achieved performance improvements by referring to these enhanced linguistic features as key-values in Transformer-based segmentation networks.

Despite these advancements, all these methods rely on the linguistic-based tokens as elements of the guidance set, as depicted in Figure 1. Since these tokens are insufficient to capture the visual contexts, these linguistic-based tokens are limited in providing the target-informative visual understanding that helps guide the network to the target areas composed of the fine-grained regions with different visual characteristics. For example, in Figure 2, the network guided by only linguistic-based tokens segments only part of the target regions (*i.e.*, 2a.A) or segments even non-target regions (*i.e.*, 2a.B). To address this issue, we explore the introduction of the *visual expression* tokens that can complement the guidance capability of linguistic information by providing the target-informative visual information.

In this paper, we propose a novel Multi-Expression guidance framework for Transformer-based Referring Image Segmentation, METRIS, which enables the network to refer to the extended guidance set composed of the visual expression as well as the linguistic expression. The proposed framework is distinct from previous studies in that we produce the visual expression tokens equipped with target-informative visual guidance capability to enhance the capacity of the guidance set and to avoid relying only on the linguistic guidance, as illustrated in Figure 1. The visual expression tokens address the lack of the guidance capacity of language-based tokens by effectively providing the visual contexts of the target regions, as shown in Figure 2a. *To the best of our knowledge, our approach is the first to generate the visual expression as a provider of the target guidance information, deviating from the previous approach in that only language-based tokens can fulfill the role of providing the target information to the network.*

Furthermore, we design a visual expression extractor from the terms of two points to generate the visual expression from vision features. To qualify as an 'expression' in this task, the following points are required: (1) It needs to concentrate more on the semantic information relevant to the target regions from the image context, because the image context contains both target and non-target information and these distracting non-target information hinders the guidance capability (Chen et al., 2024a). Thus, our module *endows with the target-oriented visual guidance ability* by selectively exploiting the informative visual tokens and adaptively refining the curated visual information. (2) It needs to capture rich visual contexts of the target regions. For this, our module considers both comprehensive context and distinct attribute contexts by exploiting the global-local linguistic cues (*i.e.*, sentence-level and word-level cues), where each of linguistic cues has different contextual information, and allows to acquire the relationship between each visual token. This design strengthens the model's adaptability to diverse language and image inputs for robust segmentation, and improves the visual understanding of the fine-grained target regions.

Our METRIS's effectiveness is clearly demonstrated by extensive experiments across multiple RIS benchmark datasets. Notably, in comparison to the ablation model using only enhanced linguistic

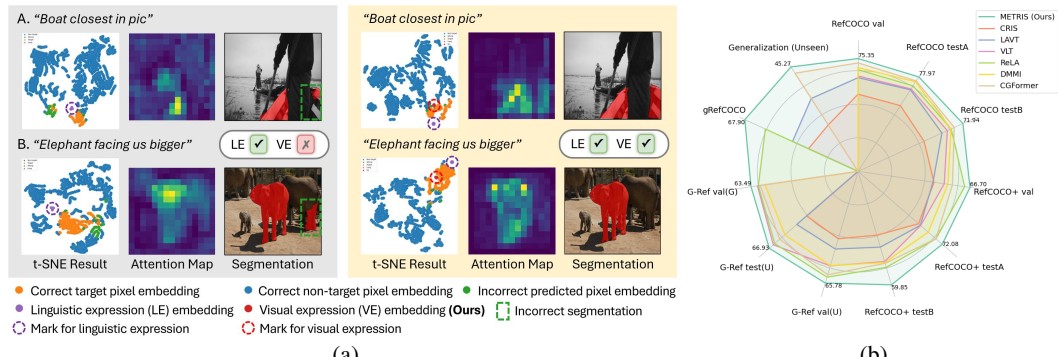

Figure 2: (a) Visual comparison of an ablation method (gray box) and our method (yellow box). In t-SNE results, the VE embedding helps to better cluster target pixel embeddings, whereas the LE embedding of the ablation method cannot sufficiently cluster target pixel embeddings. In the attention weights between the pixels and the guidance tokens, our method highlights the target regions, whereas the ablation method fails to accurately focus on target regions. In segmentation results, the ablation method guides the network to segment only some part of target regions (*i.e.*, a part of the boat) or segment even non-target regions (*i.e.*, other elephant's leg). In contrast, our method shows robust segmentation by effectively providing target-informative visual guidance. (b) Performance comparison with existing methods on a broad range of RIS benchmarks.

features as guidance elements, METRIS shows significant improvements by 3.25% oIoU on G-Ref, the most challenging dataset. In addition, our method surpasses the existing transformer-based methods on three public RIS benchmarks. As displayed in Figure 2b, we further validate the generalizability of our framework on the generalized RIS settings (Tang et al., 2023; Liu et al., 2023a). Compared to the existing state-of-the-art methods, METRIS shows stronger generalizability thanks to the introduction of the target-oriented visual guidance.

In a nutshell, our contributions can be summarized in three-fold:

- We propose METRIS, a novel Multi-Expression guidance framework for Transformer-based Referring Image Segmentation, which enables the introduction of visual expression as elements of the guidance set alongside linguistic expression to enhance the robustness of the guidance set. The visual expression addresses the lack of the guidance capacity of linguistic information by effectively providing the target-informative visual contexts. Our approach is the first to explore the potential of the visual expression as a provider of target guidance information in Transformer-based referring image segmentation.

- To produce semantic visual expression, we present a visual expression extractor designed to endow with target-oriented visual guidance ability and to capture rich visual contexts of the fine-grained target regions, thereby enhancing adaptability to diverse scenarios.

- We extensively validate our approach across the commonly used RIS settings and the generalizability evaluation settings, demonstrating the effectiveness of our framework for Transformer-based referring image segmentation. Our method consistently shows strong performance and surpasses the state-of-the-art methods on three public RIS benchmarks.

## 2 RELATED WORKS

**Transformer-based Referring Image Segmentation.** Unlike the single modal segmentation (Shim et al., 2023; Kang et al., 2024) based on fixed categories, the referring image segmentation addresses the unrestricted language expressions. Recent advanced studies have explored Transformer-based architectures that refer to the guidance information as key-value pairs, achieving great performance in this task. These studies exploited various guidance elements to guide the network to the target regions. LAVT (Yang et al., 2022), CRIS (Wang et al., 2022), VG-LAW (Su et al., 2023) used the pure linguistic features as the elements of the guidance set. LQMFormer (Shah et al., 2024) utilized learnable tokens as guidance elements, which is fine-tuned based on the language expression, to extract diverse linguistic representations. Several methods (Kim et al., 2022; Ding et al., 2022a; Hu

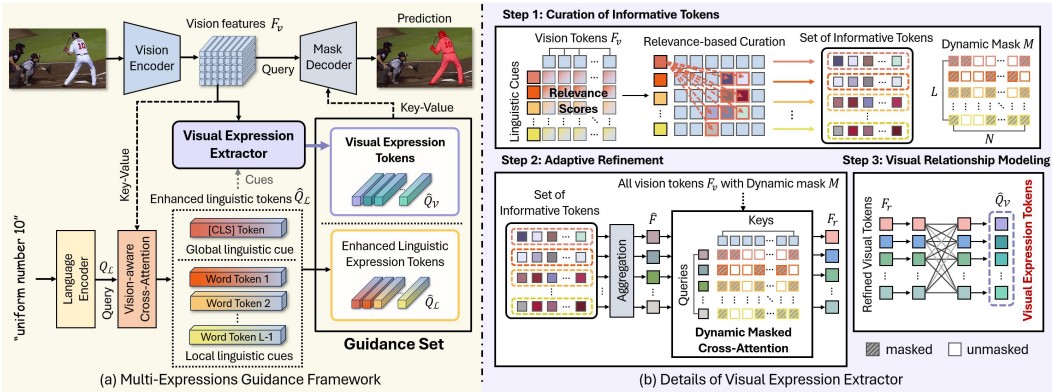

Figure 3: **Overview of METRIS.** Our approach improves the robustness of the guidance capacity via the introduction of *visual expression*. The visual expression extractor endows with the target-informative visual guidance capability via the curation of informative tokens, the adaptive refinement, and the visual relationship modeling.

et al., 2023; Tang et al., 2023; Xu et al., 2023; Wang et al., 2024) exploited the visual-attended linguistic features as the guidance elements, which are enhanced by referring to the vision features, to improve the comprehension of the language expression. More recent studies (Lai et al., 2024; Ren et al., 2024) employed the large language model (LLM) to further enhance the language understanding. LISA (Lai et al., 2024) was the first model to utilize the special linguistic token (*i.e.*, [SEG] token) generated by the multimodal LLM as the guidance element. After the success of LISA, (Ren et al., 2024; Rasheed et al., 2024; Xia et al., 2024) leveraged multiple special tokens generated by LLM as guidance elements.

Different from previous approaches, our framework exploits not only the enhanced linguistic expression tokens but also the visual expression tokens as the elements of the guidance set to avoid relying on the linguistic guidance for Transformer-based RIS. The target-informative visual guidance complements the capacity of linguistic guidance by effectively providing the visual contexts of the fine-grained target regions.

## 3 METHOD

We propose a novel multi-expression guidance framework for Transformer-based referring image segmentation, METRIS, to avoid relying on linguistic guidance. Figure 3 shows the overall framework. We first describe the vision and language feature extraction (Sec.3.1), and then introduce a visual expression extractor (Sec.3.2). Finally, we explain a segmentation decoder (Sec.3.3).

### 3.1 VISION AND LANGUAGE FEATURE EXTRACTION

Given an image $\mathcal{I}$ and a linguistic expression $\mathcal{Q}$ that consists of $L-1$ words, a vision encoder extracts the vision features $F_i \in \mathbb{R}^{H_i W_i \times C_i}$ at each stage $i \in \{1, 2, 3, 4\}$ and a language encoder extracts the linguistic expression tokens $Q_{\mathcal{L}} = [\mathbf{q}_{cls}, \mathbf{q}_1, ..., \mathbf{q}_{L-1}] \in \mathbb{R}^{L \times D}$. Note that $H_i$, $W_i$, $C_i$ and $D$ denote the height, width, channel dimension of the feature maps at the $i^{th}$ vision stage, and the channel dimension of linguistic features. The first token $\mathbf{q}_{cls}$ of linguistic expression features indicates a special [CLS] token, which is the global representation that understands the linguistic expression at the sentence level. The word token $\mathbf{q}_j$ indicates the local representation of $j^{th}$ word.

### 3.2 VISUAL EXPRESSION EXTRACTOR

To improve guidance capability, we produce the visual expression that contains target-oriented visual contexts. As shown in Figure 3 (b), the visual expression extractor consists of three steps.

**Curation of informative tokens.** This step leverages the global-local linguistic cues to consider both comprehensive context and distinct attribute contexts for rich contextual information, as each

linguistic cue captures the different contextual embedding. In this step, the linguistic expression tokens are first enhanced by the cross-attention layers using the vision features as key-value pairs to improve the comprehension for the language contexts. Then, the vision features $F_v(= F_4) \in \mathbb{R}^{N \times C}$ and the enhanced global-local linguistic tokens $\widehat{Q}_{\mathcal{L}}$ are embedded into the joint embedding space by the linear projection $\phi$, where $N$ is the total number of pixels. This process is formulated as follows:

$$X = \phi^{\mathcal{V}}(F_v) , \ Y = \phi^{\mathcal{L}}(\widehat{Q}_{\mathcal{L}}) , \tag{1}$$

After that, the relevance score map $S_c \in \mathbb{R}^{L \times N}$ between the vision tokens and the linguistic tokens is computed to curate the informative vision tokens based on linguistic cues as follows:

$$S_c = \mathcal{C}(X, Y), \ E = \mathcal{R}(S_c, r), \tag{2}$$

$$n \in \{1, 2, ..., N\}, \ l \in \{1, 2, ..., L\}, \ M_n^l = \begin{cases} 0 & n \in E^l \\ -\infty & n \notin E^l \end{cases}, \tag{3}$$

where $\mathcal{C}$ and $\mathcal{R}$ denote the cosine similarity function and the relevance-based curating operation that curates the $r$ ratio of the total vision tokens based on the higher relevance scores per linguistic cue. $\mathbb{E} \in \mathbb{R}^{L \times N_p}$ is the set of the curated token index lists per linguistic token, where $N_p$ denotes the number of the curated tokens. $M \in \mathbb{R}^{L \times N}$ is the dynamic mask that masks the non-curated tokens. As shown in Figure 3 (b), the set of informative vision tokens and the dynamic mask $M$ are passed to the adaptive refinement step.

To prevent the high relevance scores between the linguistic cues and the incorrect regions, the relevance score map $\mathbf{s} \in \mathbb{R}^{1 \times N}$ of the global linguistic token is supervised by a pixel contrastive loss:

$$\mathcal{L}_{cl} = \begin{cases} -\log(\sigma(\mathbf{s}_z/\tau)) & if \ z \in \mathcal{Z}^+ \\ -\log(1 - \sigma(\mathbf{s}_z/\tau)) & if \ z \in \mathcal{Z}^- \end{cases}, \tag{4}$$

where $\mathcal{Z}^+$ and $\mathcal{Z}^-$ denote the set of the relevant pixels and irrelevant pixels for the ground truth target regions. $\tau$ is a learnable temperature, and $\sigma$ is a sigmoid function. The pixel contrastive loss (Wang et al., 2022) encourages that the relevant pixels are embedded closer together for high relevance score and the irrelevant pixels are embedded far apart for low relevance score.

**Adaptive refinement.** Rather than simply aggregating the curated information, adaptively capturing semantic information from the curated information is more effective in producing semantic visual expression tokens. In this step, the aggregated visual tokens $F_a \in \mathbb{R}^{L \times D}$ are first obtained as:

$$S_{norm} = \texttt{Reshape}(\texttt{softmax}(S_c + M)), \ F_a = \frac{1}{N_p} \sum^{N_p} (S_{norm} \odot \texttt{Repeat}(F_v, L)), \tag{5}$$

where $\odot$ is the element-wise multiplication, and $\texttt{Repeat}(f, x)$ indicates repeating the $f$ feature $x$ times to expand the shape. The normalized score map $S_{norm} \in \mathbb{R}^{L \times N \times 1}$ is obtained by normalizing the whole relevance score map $S_c$ combined with the dynamic mask $M$. The informative visual information per linguistic cue is aggregated by the normalized weighted sum to obtain $F_a$.

Then, the refined visual tokens $F_r \in \mathbb{R}^{L \times D}$ are extracted by refining each aggregated visual token $F_a$ via the dynamic masked cross-attention mechanism to adaptively highlight the semantic information from the informative visual tokens, as follows:

$$\widehat{F} = \texttt{MHCA}(F_a, F_v, M) + F_a, \ F_r = \texttt{MLP}(\widehat{F}) + \widehat{F}, \tag{6}$$

where $\texttt{MHCA}(q, kv, m)$ denotes the multi-head cross-attention using $q$ as queries, $kv$ as key-value pairs and $m$ as masks. $\widehat{F}$ is the intermediate features. By using the dynamic mask in the masked cross-attention, the intermediate visual token $\widehat{F}$ per linguistic cue can capture semantic visual information from the informative visual tokens curated by the corresponding linguistic cue.

**Visual relationship modeling.** The visual expression tokens $\widehat{Q}_{\mathcal{V}} = [\mathbf{v}_{cls}, \mathbf{v}_1, ..., \mathbf{v}_{L-1}] \in \mathbb{R}^{L \times D}$ are produced by considering the visual relationship to mutually complement each visual token's information and acquire the visual contextual information, improving the visual understanding of the fine-grained target regions, formulated as:

$$\widehat{Q} = \texttt{MHSA}(F_r) + F_r , \ \widehat{Q}_{\mathcal{V}} = \texttt{MLP}(\widehat{Q}) + \widehat{Q} , \tag{7}$$

where $\texttt{MHSA}$ and $\widehat{Q}$ indicate the multi-head self-attention, and the intermediate features, respectively. In this way, the visual expression is endowed with the target-oriented visual guidance ability, which complements the linguistic guidance.

| | Method | Vision Encoder | Language Model | RefCOCO (Easy) | | | RefCOCO+ (Medium) | | | G-Ref (Hard) | | |
|---|---|---|---|---|---|---|---|---|---|---|---|---|
| | | | | val | test A | test B | val | test A | test B | $val_{(U)}$ | $test_{(U)}$ | $val_{(G)}$ |
| **mIoU** | CRIS (Wang et al., 2022) | CLIP R101 | CLIP | 70.47 | 73.18 | 66.10 | 62.27 | 68.08 | 53.60 | 59.87 | 60.36 | - |
| | ETRIS (Xu et al., 2023) | CLIP ViT-B | CLIP | 70.51 | 73.51 | 66.63 | 60.10 | 66.89 | 50.17 | 59.82 | 59.91 | 57.88 |
| | BarLeRIa (Wang et al., 2024) | CLIP ViT-B | CLIP | 72.4 | 75.9 | 68.3 | 65.0 | 70.8 | 56.9 | 63.4 | 63.8 | 61.6 |
| | VG-LAW (Su et al., 2023) | ViT-B | BERT-base | 75.05 | 77.36 | 71.69 | 66.61 | 70.30 | 58.14 | 65.36 | 65.13 | - |
| | PVD (Cheng et al., 2024) | Swin-B | BERT-base | 75.07 | 77.29 | 70.13 | 64.39 | 69.15 | 57.19 | 63.22 | 63.89 | 61.74 |
| | **METRIS (Ours)** | Swin-B | BERT-base | **76.97** | **78.89** | **73.63** | **68.63** | **73.88** | **61.94** | **67.85** | **67.97** | **65.86** |
| **oIoU** | LISA-7B (Lai et al., 2024) | SAM-H | LLaVA-7B | 74.1 | 76.5 | 71.1 | 62.4 | 67.4 | 56.5 | 66.4 | 68.5 | - |
| | PixelLM (Ren et al., 2024) | CLIP-VIT-L | LLaVA-7B | 73.0 | 76.5 | 68.2 | 66.3 | 71.7 | 58.3 | 69.3 | 70.5 | - |
| | SAM4MLLM-7B (Chen et al., 2025) | SAM-XL | Qwen-VL-7B-Chat | 76.2 | 80.1 | 72.0 | 71.2 | 75.9 | 64.3 | 74.2 | 74.3 | - |
| | ReSTR (Kim et al., 2022) | ViT-B | Transformer | 67.22 | 69.30 | 64.45 | 55.78 | 60.44 | 48.27 | 54.48 | - | - |
| | LAVT (Yang et al., 2022) | Swin-B | BERT-base | 72.73 | 75.82 | 68.79 | 62.14 | 68.38 | 55.10 | 61.24 | 62.09 | - |
| | VLT (Ding et al., 2022a) | Swin-B | BERT-base | 72.96 | 75.96 | 69.60 | 63.53 | 68.43 | 56.92 | 63.49 | 66.22 | 62.80 |
| | ReLA (Liu et al., 2023a) | Swin-B | BERT-base | 73.82 | 76.48 | 70.18 | 66.04 | 71.02 | 57.65 | 65.00 | 65.97 | 62.70 |
| | SADLR (Yang et al., 2023) | Swin-B | BERT-base | 74.24 | 76.25 | 70.06 | 64.28 | 69.09 | 55.19 | 63.60 | 63.56 | 61.16 |
| | DMMI (Hu et al., 2023) | Swin-B | BERT-base | 74.13 | 77.13 | 70.16 | 63.98 | 69.73 | 57.03 | 63.46 | 64.19 | 61.98 |
| | LQMFormer (Shah et al., 2024) | Swin-B | BERT-base | 74.16 | 76.82 | 71.04 | 65.91 | 71.84 | 57.59 | 64.73 | 66.04 | 62.97 |
| | CGFormer (Tang et al., 2023) | Swin-B | BERT-base | 74.75 | 77.30 | 70.64 | 64.54 | 71.00 | 57.14 | 64.68 | 65.09 | 62.51 |
| | MagNet (Cheng et al., 2024) | Swin-B | BERT-base | 75.24 | 78.24 | 71.05 | 66.16 | 71.32 | 58.14 | 65.36 | 66.03 | 63.13 |
| | **METRIS (Ours)** | Swin-B | BERT-base | 75.35 | 77.97 | 71.94 | 66.70 | 72.08 | 59.85 | 65.78 | 66.93 | 63.49 |

Table 1: Performance comparison with the state-of-the-art methods on three public referring image segmentation datasets. (U): UMD split. (G): Google split. LLM-based models are marked in gray.

## 3.3 Segmentation Decoder

To segment the target region, the decoder leverages the guidance set $\mathcal{G} = \{\widehat{Q}_{\mathcal{L}}, \widehat{Q}_{\mathcal{V}}\}$ composed of the enhanced linguistic expression tokens and the visual expression tokens. The decoder can focus its attention on more precise target regions thanks to the target-informative visual guidance. At each decoder stage, the cross-attention layer, which uses the vision features as the query and the guidance tokens as the key-value, is employed to highlight the target regions. The vision decoder features are then upsampled and concatenated with the corresponding vision encoder features to feed into the next decoder stage. The final segmentation map is projected to a binary class mask by a linear projection layer. The binary cross-entropy loss is used for the network training.

## 4 Experiments

### 4.1 Implementation Details

**Experimental settings.** The vision encoder is Swin-B (Liu et al., 2021) initialized with the pretrained weight on ImageNet-22K (Krizhevsky et al., 2012), and the language encoder is BERT-base (Devlin et al., 2018) initialized with the official pre-trained weight of the uncased version. The decoder was randomly initialized. We trained models for 40 epochs with 16 batch size on 24G RTX3090 GPUs. More details for settings are in Appendix A.

**Datasets.** RefCOCO (Yu et al., 2016) and RefCOCO+ (Yu et al., 2016) are widely utilized datasets for referring image segmentation. RefCOCO contains 19,994 images with 142,209 language expressions for 50,000 objects, and RefCOCO+ contains 19,992 images with 141,564 expressions for 49,856 objects. The expressions in RefCOCO+ do not include words about absolute locations, which makes it more challenging than RefCOCO. For RefCOCO and RefCOCO+, the target object category of the testA subset is mostly a person, and the target object of the testB subset consists of all other object categories. G-Ref (Mao et al., 2016; Nagaraja et al., 2016) is also a commonly used dataset, which contains 26,711 images with 104,560 language expressions for 54,822 objects. G-Ref, which is the most challenging dataset, has more complex and longer expressions than RefCOCO and RefCOCO+.

**Evaluation metrics.** Following previous works, we adopted the overall intersection-over-union (oIoU), mean intersection-over-union (mIoU), and precision at 0.5, 0.7 and 0.9 thresholds.

### 4.2 Comparison with State-of-The-Art Transformer-based RIS Methods

In Table 1, we evaluated our approach with Transformer-based RIS methods on three public benchmarks. Our method consistently showed strong performance on all evaluation splits of all datasets, whereas previous methods usually overfit to some evaluation splits.

| Method | val(U) | | test(U) | | val(G) | |
|---|---|---|---|---|---|---|
| | seen | unseen | seen | unseen | seen | unseen |
| CRIS | 58.64 | 42.63 | 59.68 | 38.88 | 42.36 | 32.84 |
| LAVT | 60.16 | 42.33 | 60.37 | 41.38 | 57.33 | 40.43 |
| CGFormer | 65.60 | 46.11 | 65.67 | 42.31 | 62.85 | 45.05 |
| **METRIS** | **66.52** | **46.74** | **66.93** | **43.06** | **63.61** | **46.01** |

Table 3: Comparison for generalization setting on G-Ref using mIoU.

| Method | val | | testA | | testB | |
|---|---|---|---|---|---|---|
| | mIoU | oIoU | mIoU | oIoU | mIoU | oIoU |
| CRIS (Wang et al., 2022) | 56.27 | 55.34 | 63.42 | 63.82 | 51.79 | 51.04 |
| LAVT (Yang et al., 2022) | 58.40 | 57.64 | 65.90 | 65.32 | 55.83 | 55.04 |
| ReLA (Liu et al., 2023a) | 63.60 | 62.42 | 70.03 | 69.26 | 61.02 | 59.88 |
| GSVA-7B (Xia et al., 2024) | 66.47 | 63.29 | 71.08 | 69.93 | 62.23 | 60.47 |
| **METRIS** | **69.37** | **65.88** | **72.81** | **71.74** | **64.29** | **63.30** |

Table 4: Comparison with previous methods on gRefCOCO. Gray is a LLM-based model.

performance compared to the recent state-of-the-art methods such as DMMI, LQMFormer and CGFormer, which leverage the enhanced linguistic tokens as the guidance elements. Furthermore, as shown in Table 2, METRIS showed higher mIoU and oIoU performance with comparable computations to DMMI and with 45.5% less computations than CGFormer on the most challenging dataset. These results demonstrate the effectiveness of our approach.

| Method | MACs | G-Ref val(U) | |
|---|---|---|---|
| | | mIoU | oIoU |
| DMMI | 392 G | 66.48 | 63.46 |
| CGFormer | 950 G | 67.57 | 64.68 |
| **METRIS** | 432 G | **67.85** | **65.78** |

Table 2: Computational cost (MACs) and performance comparison.

In addition, we validated the generalizability of our framework compared to other methods. In this task, the ability to understand the visual context within the image is particularly important for improving generalizability. In Table 3, we experimented with the generalization setting (Tang et al., 2023), where only the language descriptions for the seen target object classes are given during training and the model is not trained with the ground truth masks for the unseen target object classes. METRIS surpassed the existing methods and consistently showed performance improvements on both seen and unseen sets. In Table 4, we experimented on the generalized RIS benchmark (gRefCOCO) (Liu et al., 2023a) that includes more comprehensive scenarios such as multi-target and no-target samples. Compared to ReLA, METRIS showed remarkable improvements by 3.46%, 1.71% and 3.42% oIoU on each split, respectively. These results suggest that our method has a better generalization ability than previous RIS methods in this task by learning a wider variety of the visual contexts via the visual expression.

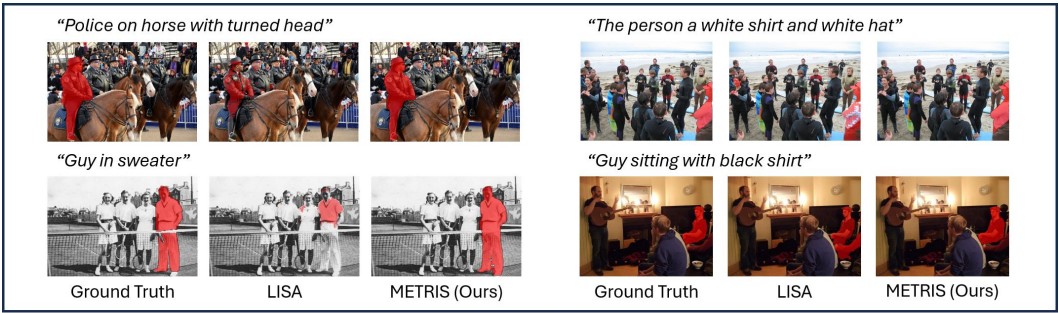

Figure 4: Qualitative comparison with the LLM-based RIS model (Lai et al., 2024) on RefCOCO+.

## 4.3 COMPARISON WITH LLM-BASED RIS METHODS

Despite the unfair comparison, we conducted comparison with the LLM-based RIS models in Table 1 for further analysis. Our model showed competitive performance without the LLM's ability on three benchmarks. Furthermore, we compared segmentation results in Figure 4. Our model showed accurate segmentation, whereas LISA segmented only some part of a target object or segment even non-target regions. These results indicate that our model has a stronger ability to understand the visual contexts of the target regions compared to the LLM-based model, which relies on the generated linguistic token.

## 4.4 ABLATION STUDIES

All ablation models are based on our network. For a fair comparison, we added the cross-attention layers into the ablation models to maintain the model size similar to our default model.

| Guidance Element | | RefCOCO *val* (Easy) | | | | | RefCOCO+ *val* (Medium) | | | | | G-Ref *val*(U) (Hard) | | | | |
|---|---|---|---|---|---|---|---|---|---|---|---|---|---|---|---|---|
| *Linguistic* | *Visual* | P@0.5 | P@0.7 | P@0.9 | mIoU | oIoU | P@0.5 | P@0.7 | P@0.9 | mIoU | oIoU | P@0.5 | P@0.7 | P@0.9 | mIoU | oIoU |
| Pure LE | ✗ | 84.73 | 75.49 | 34.87 | 74.61 | 72.85 | 73.54 | 64.59 | 28.35 | 63.72 | 62.15 | 72.77 | 59.90 | 22.86 | 62.52 | 61.59 |
| Enhanced LE | ✗ | 85.46 | 76.22 | 36.04 | 75.10 | 73.56 | 74.90 | 66.12 | 29.83 | 65.46 | 63.97 | 74.02 | 61.28 | 24.55 | 64.35 | 63.68 |
| ✗ | VE | 86.38 | 77.82 | 36.90 | 75.84 | 74.52 | 76.29 | 67.60 | 31.36 | 67.33 | 65.59 | 74.89 | 63.03 | 26.33 | 66.31 | 65.45 |
| Enhanced LE | All pixels | 86.17 | 77.40 | 36.73 | 75.65 | 74.36 | 75.81 | 67.28 | 30.89 | 66.97 | 65.24 | 74.85 | 62.77 | 25.91 | 66.02 | 65.27 |
| Enhanced LE | VE | **86.71** | **78.30** | **37.24** | **76.97** | **75.35** | **77.13** | **69.05** | **32.94** | **68.63** | **66.70** | **76.13** | **64.60** | **27.87** | **67.85** | **66.93** |

Table 5: Main ablation for the effectiveness of our multi-expression guidance. LE: Linguistic Expression tokens. VE: Visual Expression tokens (Ours). ▨ are models with target-informative linguistic guidance only. ▨ is a model with target-informative visual guidance only. ▨ is a model using all visual information as visual guidance. ▨ is our full model.

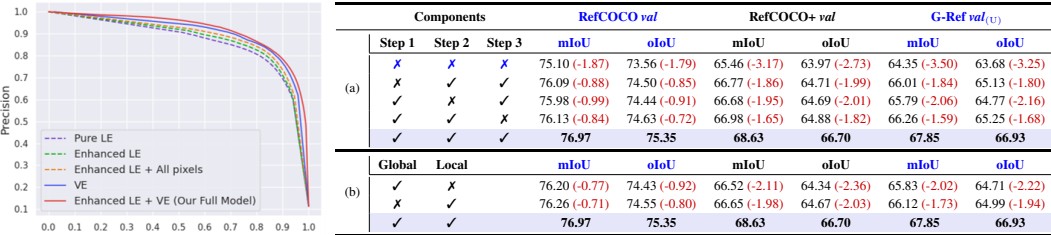

| | Components | | | RefCOCO *val* | | RefCOCO+ *val* | | G-Ref *val*(U) | |
|---|---|---|---|---|---|---|---|---|---|
| | Step 1 | Step 2 | Step 3 | mIoU | oIoU | mIoU | oIoU | mIoU | oIoU |
| (a) | ✗ | ✗ | ✗ | 75.10 (-1.87) | 73.56 (-1.79) | 65.46 (-3.17) | 63.97 (-2.73) | 64.35 (-3.50) | 63.68 (-3.25) |
| | ✗ | ✓ | ✓ | 76.09 (-0.88) | 74.50 (-0.85) | 66.77 (-1.86) | 64.71 (-1.99) | 66.01 (-1.84) | 65.13 (-1.80) |
| | ✓ | ✗ | ✓ | 75.98 (-0.99) | 74.44 (-0.91) | 66.68 (-1.95) | 64.69 (-2.01) | 65.79 (-2.06) | 64.77 (-2.16) |
| | ✓ | ✓ | ✗ | 76.13 (-0.84) | 74.63 (-0.72) | 66.98 (-1.65) | 64.88 (-1.82) | 66.26 (-1.59) | 65.25 (-1.68) |
| | ✓ | ✓ | ✓ | **76.97** | **75.35** | **68.63** | **66.70** | **67.85** | **66.93** |
| | Global | Local | | mIoU | oIoU | mIoU | oIoU | mIoU | oIoU |
| (b) | ✓ | ✗ | | 76.20 (-0.77) | 74.43 (-0.92) | 66.52 (-2.11) | 64.34 (-2.36) | 65.83 (-2.02) | 64.71 (-2.22) |
| | ✗ | ✓ | | 76.26 (-0.71) | 74.55 (-0.80) | 66.65 (-1.98) | 64.67 (-2.03) | 66.12 (-1.73) | 64.99 (-1.94) |
| | ✓ | ✓ | | **76.97** | **75.35** | **68.63** | **66.70** | **67.85** | **66.93** |

Figure 5: Precision-Recall curves of ablation models on RefCOCO+.

Table 6: Ablation studies for the design of our visual expression extractor on three public benchmarks. Our default design is marked in ▨. Drops are relative to our default design.

**Effectiveness of Target-oriented Visual Guidance.** In Table 5, we conducted experiments to validate the effectiveness of exploiting the visual expression tokens as the elements of the guidance set alongside the linguistic expression tokens. Compared to 'Pure LE' method that uses only the pure language encoder features $Q_{\mathcal{L}}$ as guidance elements, 'Enhanced LE' method (our baseline), which uses only the enhanced linguistic tokens $\widehat{Q}_{\mathcal{L}}$ as guidance elements, showed better performance on each dataset. This suggests that the enhancement of the language features by referring to the visual information helps to improve the comprehension for the meaning of the language expression context. Compared to these two methods, our full method showed remarkable improvements by 5.34% and 3.25% oIoU on G-Ref, the most challenging dataset. These results indicate that linguistic guidance capacity is insufficient to provide the visual understanding of the fine-grained target regions, and the introduction of visual expression tokens as guidance elements can effectively complement the linguistic guidance capacity.

Furthermore, 'VE only' method (row3) showed a significant increase of 1.77% oIoU than 'Enhanced LE' method on G-Ref. These interesting results demonstrated the effectiveness of the visual expression *itself*. In addition, we compared our full method with the 'all-pixel' method (row4) that uses all visual pixels as visual guidance elements. Even though the 'all-pixel' method can provide the unique visual information to the network, our method showed 1.66% higher oIoU on G-Ref. This indicates that distracting non-target visual information hinders the guidance capability. Thus, our visual expression's target-oriented visual guidance is more effective at improving the ability to understand the visual contexts of the target regions than using all of pixels.

In Figure 5, we also displayed the precision-recall curves. The area under curve (AUC-PR) summarizes the overall performance of the model across different threshold values. As shown in Figure 5, 'VE only' method maintained its advantage in precision over the 'Pure LE' and 'Enhanced LE' methods. Our full model had the highest AUC-PR.

**Analysis on Components of Visual Expression Extractor.** In Table 6, we conducted the ablation on the design of our visual expression extractor. To keep the parameter size similar for a fair comparison, we added more attention layers into the ablation models. As displayed in Table 6 (a), the removal of Step 1 resulted in 0.85%, 1.99%, and 1.80% drops in oIoU on each dataset. These results indicate that it is effective to concentrating more on the informative tokens from the image context that contains both the target-relevant information and the distracting non-target information. The removal of Step 2 decreased oIoU performance by 2.16% on G-Ref. This result highlights that adaptively capturing the semantic information from the curated information is more effective than simply aggregating the curated information for producing more semantic visual expression. The

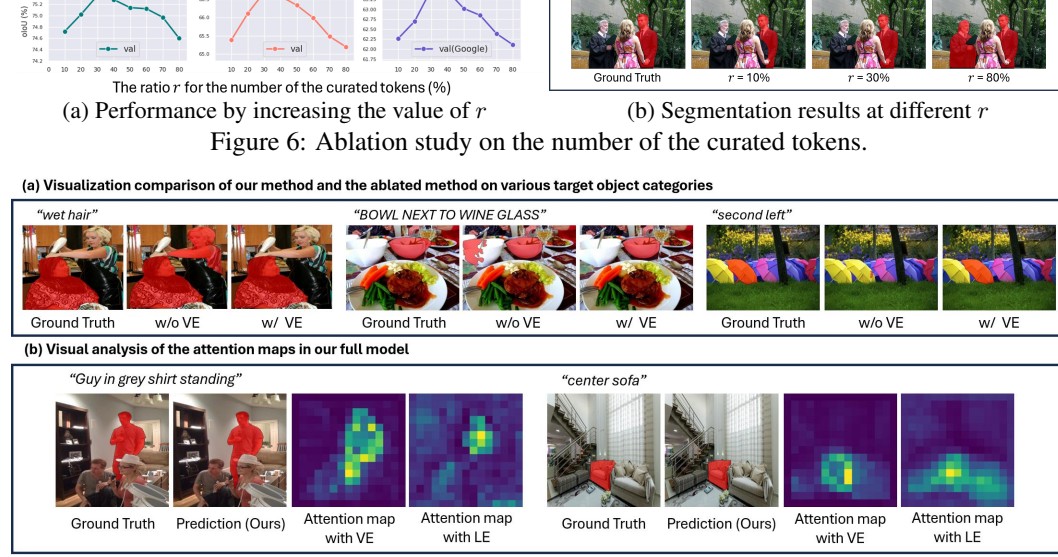

(a) Performance by increasing the value of $r$      (b) Segmentation results at different $r$

Figure 6: Ablation study on the number of the curated tokens.

Figure 7: (a) Visualization of our method and the ablated method on various target objects. (b) Visual analysis of the attention maps in our full model. More results are provided in Appendix.

removal of Step 3 resulted in a 1.82% drop in oIoU on RefCOCO+. This indicates that each token of the visual expression acquires the visual context information for target regions by considering the relationship between each visual token. These ablation studies demonstrate that each of the proposed components is necessary to endow the visual expression tokens with the target-oriented visual guidance capability.

As shown in Table 6 (b), removing the use of the local linguistic cues showed a 2.36% drop in oIoU compared to our full model on RefCOCO+. In addition, removing the use of the global linguistic cue showed a 2.03% drop in oIoU on RefCOCO+. These results demonstrated that using both global and local linguistic cues allows the visual expression tokens to consider both the comprehensive context and the distinct attribute context in order to the enriched visual contexts of the fine-grained target regions, as each of linguistic cues has different contextual information.

**Number of Curated Tokens.** We analysed the value of $r$, which is the ratio for the number of the curated tokens. Compared to the $r$ values of 10 and 80, the $r$ of 30 showed higher oIoU in Figure 6 (a). In addition, as shown in Figure 6 (b), the $r$ of 30 segmented more clearly, while the $r$ of 10 missed some part of the target regions and the $r$ of 80 even segmented other object regions. The smaller number of $k$ resulted in a lack of information, where the semantic visual information cannot be sufficiently exploited. In contrast, the larger number of $r$ resulted in including the noise information and degrades the guidance capability. Therefore, the optimal $r$ can selectively exploit the semantic visual information and filter out noise components to improve the robustness of the guidance capacity.

## 4.5 QUALITATIVE RESULTS

In addition to the visual comparisons (*i.e.*, t-SNE and attention maps) in Figure 2, we compared the segmentation results on various target object categories in Figure 7 (a). Our method consistently predicted the accurate regions by leveraging the visual expression, while the ablation method included the wide non-target regions or missed the target regions.

Furthermore, in Figure 7 (b), we displayed additional visual analysis of the attention map between the vision features and the visual expression and the attention map between the vision features and the enhanced language expression in our full model. The results showed that our visual expression complements the target information even though the enhanced language expression misses the target regions or includes even non-target regions, addressing the lack of guidance caused by the visual-aware linguistic token's limitation.

Figure 8: Visualizations for the different types of the images and language expressions on Ref-COCO+ and G-Ref.

In Figure 8 (a), we compared with previous Transformer-based RIS methods, which use only the enhanced linguistic tokens as the guidance set, on diverse types of inputs. Our method segmented more clearly for the complicated images and the ambiguous language expressions, whereas other methods incorrectly predicted and uncertainly segmented the regions. These results indicate that our approach is more effective in improving visual understanding of the target regions. In Figure 8 (b), we visualized the results on longer and more complex language expressions. These results indicate that METRIS effectively enhances the robustness of the network for the complex scenarios.

### 4.6 CONCLUSION

We propose a novel Multi-Expression guidance framework for Transformer-based Referring Image Segmentation, METRIS, which enables the introduction of the visual expression as elements of the guidance set alongside the linguistic expression to enhance the robustness of the guidance capability. Our approach explores the potential of the visual expression as a provider of target guidance information, beyond the previous approach in that only language-based tokens can fulfill the role of providing target-informative guidance information. The visual expression complements the capability of linguistic guidance by effectively providing the target-oriented visual guidance. To produce semantic visual expression, we present a visual expression extractor that is designed to endow with the target-informative visual guidance ability and to acquire the rich contextual information of target regions. This enhances the adaptability to diverse image and language inputs, and improves visual understanding of the fine-grained target regions. Extensive comparisons and ablations demonstrated the effectiveness of our approach for Transformer-based referring image segmentation.

**Limitation and Future Work.** Despite METRIS's stronger ability to understand the visual contexts of the target regions than LLM-based models, our model showed lower performance on the most challenging dataset (G-Ref), which consists of the difficult language samples. This means that our model lacks the reasoning ability for the implicit and detailed descriptions in comparison to the LLM-based models. This finding suggests that our performance bottleneck may still lie in understanding the language expressions on this task, while our model has better performance than the existing state-of-the-art Transformer-based RIS models in Table 1. Therefore, future work could have a broader impact on this task via the exploration of combining our approach's strength with the LLM's strength, beyond relying on the LLM's capability.

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

| Method | Large-scale Training Datasets | Vision Encoder | RefCOCO | | | RefCOCO+ | | | G-Ref | | |
|---|---|---|---|---|---|---|---|---|---|---|---|
| | | | val | test A | test B | val | test A | test B | val(U) | test(U) | val(G) |
| X-Decoder (B) (Zou et al., 2023) | ✓ | DaViT-B (Ding et al., 2022b) | - | - | - | - | - | - | 64.5 | - | - |
| SEEM (B) (Zou et al., 2024) | ✓ | DaViT-B (Ding et al., 2022b) | - | - | - | - | - | - | 65.0 | - | - |
| PolyFormer (Liu et al., 2023b) | ✓ | Swin-B (Liu et al., 2021) | 74.82 | 76.64 | 71.06 | **67.64** | **72.89** | 59.33 | 67.76 | 69.05 | - |
| **METRIS (Ours)** | ✗ | Swin-B | **75.35** | **77.97** | **71.94** | 66.70 | 72.08 | **59.85** | 65.78 | 66.93 | **63.49** |

Table 7: oIoU performance comparison with other RIS models, which use the additional large scale vision-language datasets at training, on three public referring image segmentation benchmarks. (U): UMD split. (G): Google split. The best score is in **bold**.

| Method | mIoU | oIoU |
|---|---|---|
| ✗ | 67.54 (-1.09) | 65.43 (-1.27) |
| ✓ | **68.63** | **66.70** |

(a) Supervised by the contrastive loss

| Method | mIoU | oIoU |
|---|---|---|
| w/o Dynamic mask | 66.91 (-1.72) | 64.95 (-1.75) |
| w/ Dynamic mask | **68.63** | **66.70** |

(b) Normalization with the dynamic mask

Table 8: Additional ablation on the detailed design choice of METRIS.

# APPENDIX

# A  ADDITIONAL IMPLEMENTATION DETAILS

**Experimental Settings.** Our method was implemented in PyTorch (Paszke et al., 2019). We used the AdamW (Loshchilov & Hutter, 2017) optimizer with initial learning rate of 3e-5 and adopted the polynomial learning rate decay scheduler. The input image resolution was 480×480. For gRef-COCO that contains no-target samples, we used a no-target classifier (Liu et al., 2023a).

**Evaluation Metrics.** Following previous works, we adopted the overall intersection-over-union (oIoU), mean intersection-over-union (mIoU), and precision at 0.5, 0.7 and 0.9 thresholds. The oIoU is the ratio between the total intersection regions and the total union regions of all test samples. The mIoU is the average of IoUs between the predicted mask and the ground truth of all test samples. The precision is the percentage of test samples that have an IoU score higher than a threshold.

# B  ADDITIONAL DETAILS FOR GENERALIZATION SETTING

To further validate the generalization ability of our model, we experimented on the generalization setting introduced by (Tang et al., 2023). These setting splits the RIS datasets into the seen and unseen categories on MSCOCO (Lin et al., 2014) of the open-vocabulary detection (Zareian et al., 2021). The training set contains GT masks for only seen categories, and the test set consists of the seen subset and the unseen subset. Following the previous work (Tang et al., 2023), we adopted the text encoder of CLIP (Radford et al., 2021) as the language encoder for a fair comparison in this experiment, and trained our model for 50 epochs.

# C  ADDITIONAL DETAILS FOR DATASETS

**RefCOCO & RefCOCO+.** These two datasets are distributed under the Apache-2.0 license, and are collected from the two-player game (Yu et al., 2016). The evaluation sets of RefCOCO and RefCOCO+ are splitted into the validation subset, the test A subset and the test B subset. The images of the testA subset contain the multiple people, and the images of the testB subset contain the multiple instances of all other objects. RefCOCO+, which forbids the words about the absolute locations in the language expressions, is more challenging than RefCOCO.

**G-Ref.** This dataset is distributed under the CC-BY 4.0 license, and is collected on Amazon Mechanical Turk. We use both UMD (Nagaraja et al., 2016) and Google (Mao et al., 2016) partitions for the evaluation. The UMD partition splits the evaluation set into the validation subset and the test subset. The Google partition consists of only the validation set. The average length of the language expressions is 8.4 words. This means that the G-Ref dataset contains longer and more complex language expressions than the RefCOCO and RefCOCO+ datasets. Thus, G-Ref is the most challenging dataset.

| Method | RefCOCO *val* | | RefCOCO+ *val* | | G-Ref *val*$_{(U)}$ | |
|---|---|---|---|---|---|---|
| | mIoU | oIoU | mIoU | oIoU | mIoU | oIoU |
| w/o articles | 76.59 (-0.38) | 74.93 (-0.42) | 68.23 (-0.40) | 66.12 (-0.58) | 67.34 (-0.51) | 66.39 (-0.54) |
| All words | **76.97** | **75.35** | **68.63** | **66.70** | **67.85** | **66.93** |

Table 9: Ablation study on the use of the article tokens at the process of collecting informative visual regions.

## D    COMPARISON TO RIS MODELS TRAINED WITH ADDITIONAL LARGE-SCALE DATASETS

To further analysis of our method, we compared our model with other RIS models (Zou et al., 2023; 2024; Liu et al., 2023b) that use the additional large scale vision-language grounding datasets (Plummer et al., 2015; Krishna et al., 2017; Chen et al., 2015) at training. Since training with multiple datasets brings the significant performance improvement on referring segmentation, Poly-Former (Liu et al., 2023b) showed higher performance on four splits (*i.e.*, RefCOCO+ *val.* & *test A*, and G-Ref *val*$_{(U)}$ & *test*$_{(U)}$). However, even though a direct comparison between our model and PolyFormer is unfair, our model outperformed PolyFormer on the other 5 splits. These results demonstrate the great adaptability of our approach.

## E    ADDITIONAL ABLATION ON DESIGN CHOICE

**Supervision by the contrastive loss.** In Table 8 (a), we experimented on supervising the relevance score map by the pixel contrastive loss (Eq.4). This result indicates that the contrastive loss helps to monitor the curation of the informative tokens associated with the correct target region and to prevent the high relevance scores between the linguistic features and incorrect regions.

**Normalization with dynamic mask.** We ablated on applying a softmax normalization with the dynamic mask to the relevance scores (Eq.5). In Table 8 (b), normalizing without the dynamic mask showed a significant performance drop. This indicates that using the curated visual tokens is beneficial for robust segmentation than using all visual tokens including the distracting tokens.

**The use of the meaningless words.** we experimented the ablation on the use of the article tokens such as "the", "a" and "an", which are meaningless words in the input sentence, in the process of collecting informative visual regions. As shown in Table 9, compared to using all word tokens, 'w/o article' resulted in 0.42%, 0.58% and 0.54% drops in oIoU on each dataset, respectively. These results indicate that the article tokens do not carry the noise information, and using all word tokens as linguistic cues are more effective at collecting the informative visual tokens. Since the relations of each word are considered during encoding the language input to capture the contextual information for the target object description, each language token is encoded with semantic representations to guide to the target object.

## F    ADDITIONAL QUALITATIVE RESULTS

As illustrated in Figure 9, we visualized additional results of our full model and the ablation model for two or three different language expressions describing the same object. Our method showed robust segmentation for various language expressions, whereas the ablation model segmented the non-target regions or did not highlight the target regions. In addition, we displayed additional qualitative results on various scenarios in Figure 10 and Figures 11 to 14. Furthermore, we showed additional visual analysis of the attention map between the vision features and the visual expression in comparison to the attention map between the vision features and the enhanced language expression in our full model. As shown in Figure 15, the visual expression addressed the regions where the enhanced language expression includes despite of the non-target regions or fails to highlight.

**(a) Three different language expressions**

"second guy from right"  "man with scarf"  "guy with scarf"

"yellow shirt by the ballerina"  "kid on left, yellow shirt"  "player left side"

Image    Ground Truth    w/o visual expression   METRIS (Ours)    w/o visual expression   METRIS (Ours)    w/o visual expression   METRIS (Ours)

**(b) Two different language expressions**

"left smiling man"  "standing man on left"

"right half"  "right half of pizza"

"man on left kneeling"  "guy signing shirt"

Image    Ground Truth    w/o visual expression   METRIS (Ours)    w/o visual expression   METRIS (Ours)

Figure 9: Additional qualitative comparison of the proposed method and the ablated model on different language expressions describing the same object in the image.

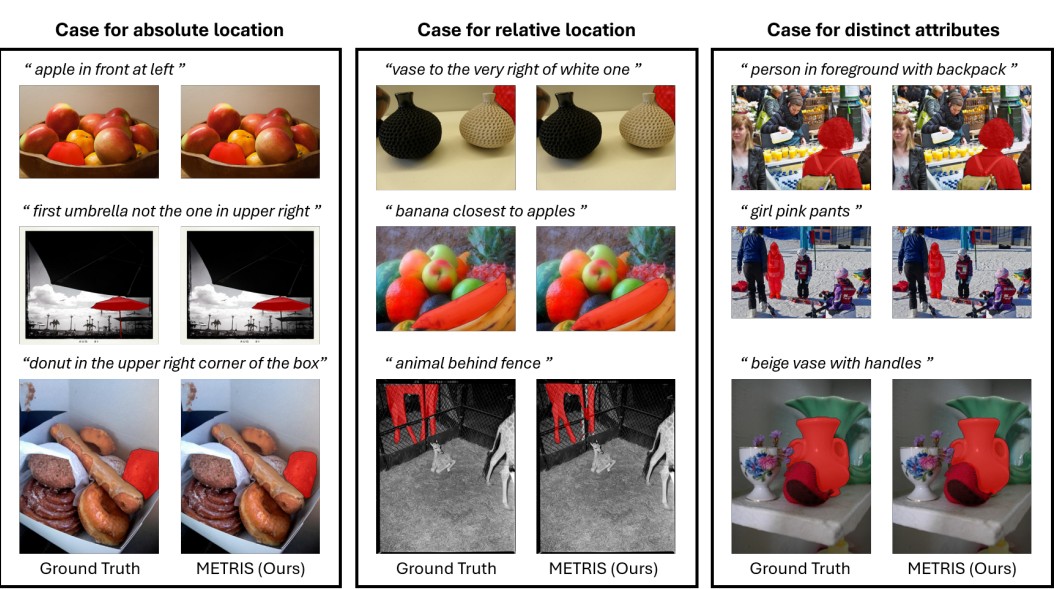

Figure 10: Additional qualitative results on more diverse language expressions and images.

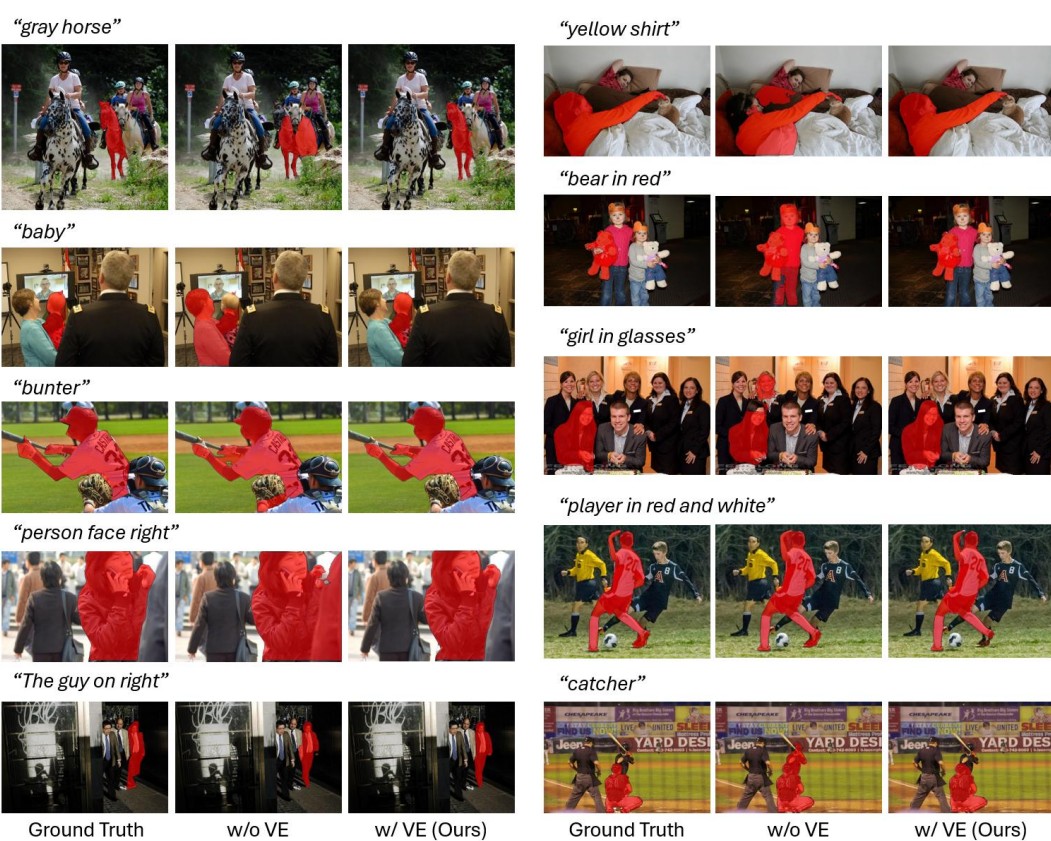

Figure 11: Visualization comparison of our method and the ablated method on the target regions of the person, where the ablation model without the visual expression segments even non-target regions.

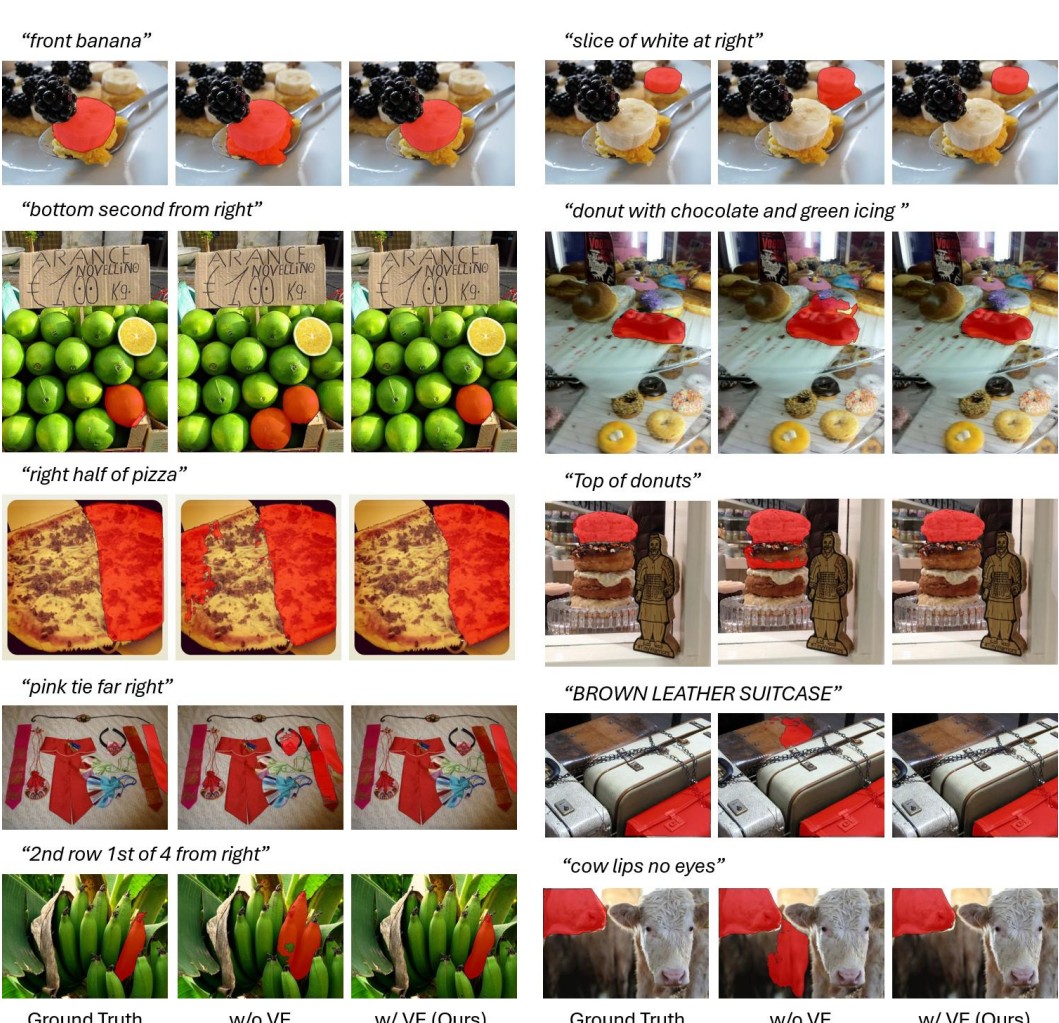

*"front banana"*

*"slice of white at right"*

*"bottom second from right"*

*"donut with chocolate and green icing "*

*"right half of pizza"*

*"Top of donuts"*

*"pink tie far right"*

*"BROWN LEATHER SUITCASE"*

*"2nd row 1st of 4 from right"*

*"cow lips no eyes"*

Ground Truth     w/o VE     w/ VE (Ours)       Ground Truth     w/o VE     w/ VE (Ours)

Figure 12: Visualization comparison of our method and the ablated method on various target object categories, where the ablation model without the visual expression segments even non-target regions.

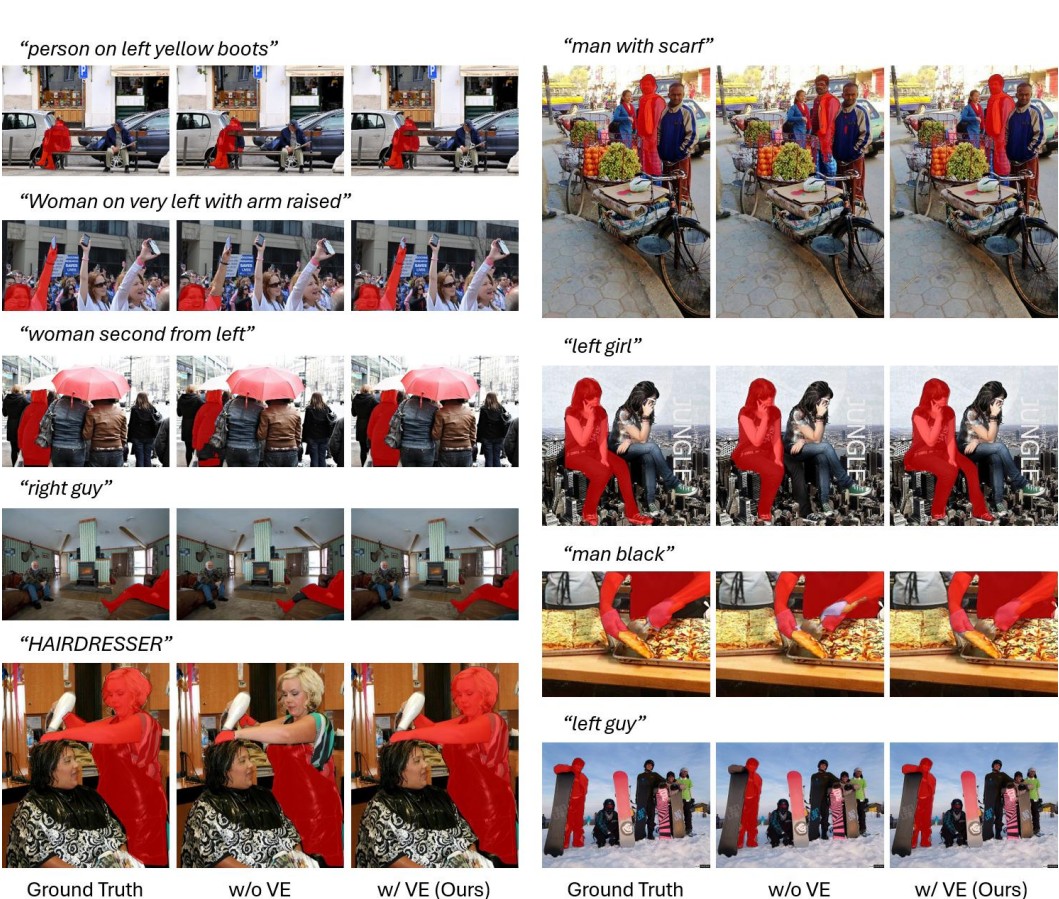

Figure 13: Visualization comparison of our method and the ablated method on the target regions of the person, where the ablation model without the visual expression fails to capture the target regions.

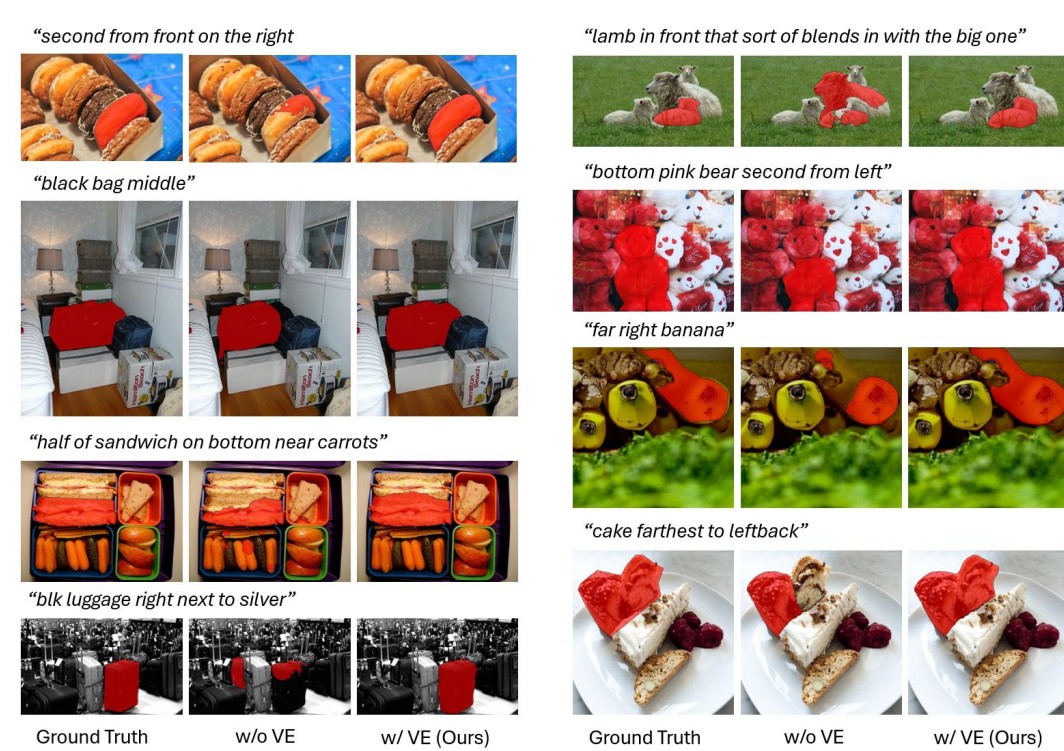

Figure 14: Visualization comparison of our method and the ablated method on various target object categories, where the ablation model without the visual expression fails to capture the target regions.

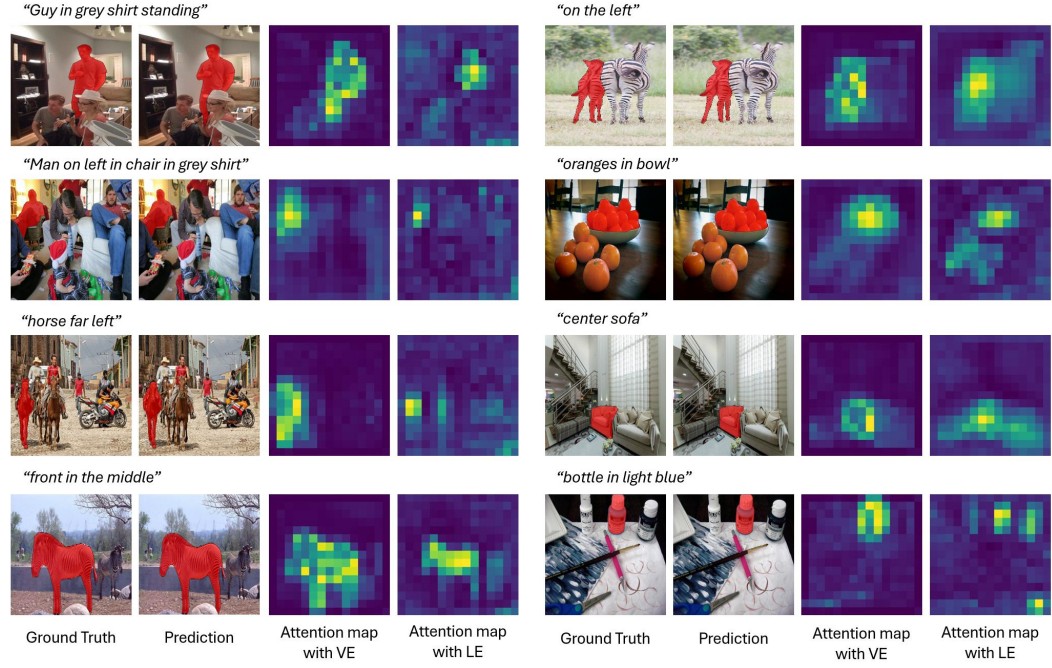

Figure 15: Visual analysis of the attention map between the vision features and the visual expression and the attention map between the vision features and the enhanced language expression. The prediction results are predicted by our full model.

