# OpenReview forum: "METRIS: Multi-Expressions for Transformer-based Referring Image Segmentation"
_ICLR.cc/2025/Conference — Submitted to ICLR 2025_

### Official Review · Reviewer_6fUF · 2024-10-31

**Soundness:** 3
**Presentation:** 3
**Contribution:** 3
**Rating:** 6
**Confidence:** 4

**Summary:**

This paper introduces a Multi-Expression guidance framework for Referring Image Segmentation (RIS) that enhances Transformer-based networks. Unlike traditional RIS methods that rely solely on linguistic tokens for guidance, METRIS overcomes this limitation by incorporating visual expression tokens alongside linguistic ones, enabling the network to access more informative visual contexts. A visual expression extractor is introduced to generate semantic visual expressions, enhancing adaptability to diverse image and language inputs. Extensive experiments demonstrate that METRIS improves visual understanding and consistently outperforms state-of-the-art methods across three public RIS benchmarks.

**Strengths:**

- This paper produces the target-informative visual expression for Transformer-based referring image segmentation.
- This paper presents a visual expression extractor designed to provide target-oriented visual guidance and capture rich visual contexts of fine-grained target regions, thereby enhancing adaptability to diverse scenarios.
- Experiments on three public RIS benchmarks demonstrate its effectiveness.

**Weaknesses:**

- I'm a bit confused about the motivation behind the paper. The task involves segmenting the target object area given a query, but the paper combines target information with text input. Doesn’t this amount to leaking explicit prior information—essentially guiding the model to find the corresponding location in the image? This seems to weaken the inherent difficulty of the task.
- Collecting informative visual regions based on all word tokens, including meaningless ones like "the" and "an," seems inappropriate. These tokens do not carry relevant information for identifying the target object and may introduce noise into the process. It would be more effective to focus on meaningful tokens that contribute to understanding the target context.
- It seems that the operations in the paper are quite complex. How does the complexity of this method compare to existing approaches?

**Questions:**

- I'm a bit confused about the motivation behind the paper. The task involves segmenting the target object area given a query, but the paper combines target information with text input. Doesn’t this amount to leaking explicit prior information—essentially guiding the model to find the corresponding location in the image? This seems to weaken the inherent difficulty of the task.
- Collecting informative visual regions based on all word tokens, including meaningless ones like "the" and "an," seems inappropriate. These tokens do not carry relevant information for identifying the target object and may introduce noise into the process. It would be more effective to focus on meaningful tokens that contribute to understanding the target context.
- It seems that the operations in the paper are quite complex. How does the complexity of this method compare to existing approaches?

---

> ### Author Response · Authors · 2024-11-20
> **Response to Reviewer 6fUF (1/n)**
>
> > **W1. Concern about leaking prior information**
>
> We sincerely appreciate the reviewer’s constructive comment, which allowed us to clarify our method with a new perspective. As the reviewer mentioned, this task aims to segment the target region given a language expression that describes the target object. However, the language expression has a limitation due to the insufficiency of target information, as shown in Figure 2.
>
> To address this limitation, we leverage the similarity-based visual information retrieval based on the linguistic cues about the target object, and exploit both the retrieved visual information (i.e., target-informative visual tokens) and the linguistic target information to enhance the target-informative guidance capability. During generating the visual expression, we did not use any labeled information (e.g., ground truth mask), which may cause the leaking of prior information. Therefore, we believe that *this approach could not be considered as a prior information leakage, but rather could be considered as the retrieval of visual information to address the lack of linguistic guidance capability.* (We guessed that the omission of a notation for linguistic cues in Figure 1 of our draft may have confused the reviewer, so we modified Figure 1 of the revised paper by adding a notation of linguistic cues.)
>
> Furthermore, upon analyzing our visual expression, we found that it contained semantic information that is highly relevant to the target regions. This is the reason why we called the additional visual information as a target-informative visual expression, which significantly complements the linguistic guidance capability.
>
> To provide the in-depth understanding and further clarify the effectiveness of our approach, we visualized the attention maps of the language expression and the visual expression, respectively, in Figures 7 and 15 of the revised paper. The results show that the visual expression captures the semantic information for the target regions missed by the language expression.
>
> We hope that our explanation addresses the reviewer’s concerns, and we sincerely thank the reviewer for the thoughtful comment, which is valuable in further clarifying the potentially confusing aspects of our paper.

---

> ### Author Response · Authors · 2024-11-20
> **Response to Reviewer 6fUF (2/n)**
>
> >**W2. Addressing the impact of article words  (e.g., "a", "an", and "the")**
>
> Thank you for the reviewer’s insightful feedback. As the reviewer suggested, we conducted the ablation experiment with excluding the use of article tokens, such as “the”, “a” and “an”, in the process of collecting informative visual regions. As shown in Table L, compared to using all word tokens, ‘w/o article’ resulted in 0.42%, 0.58% and 0.54% drops in oIoU on each dataset, respectively. These results indicate that the article tokens do not carry the noise information. Since the relations of each word are considered during encoding the language input to capture the contextual information for the target object description, each language token is encoded with semantic representations to guide to the target object. We added this ablation results in Table 9 of the revised paper, and we hope this experiment addresses the reviewer's concern.
>
> >**Table L. Ablation study on the use of the article tokens as the linguistic cues**
> | Method  | RefCOCO val  mIoU / oIoU | RefCOCO+ val  mIoU / oIoU | G-Ref val  mIoU / oIoU |
> |----------|------------|------------|------------|
> | w/o articles   | 76.59      / 74.93       | 68.23   / 66.12    | 67.34     / 66.39   |
> | All words     | **76.97**   / **75.35**   | **68.63**   /  **66.70**    | **67.85** / **66.93**   |
>
> ---
> >**W3. Complexity comparison**
>
> Thanks for the reviewer's valuable comment regarding the complexity of our approach. As shown in Table M, we compared the computational complexity (MACs) of our method and the existing methods. Our model showed a significant improvement of 2.03% on average with a comparable complexity to DMMI on all datasets. In addition, our model showed on average of 1.38% higher performance despite having 45.5% less complexity compared to CGFormer. We hope our response clarifies the complexity concerns.
>
> >**Table M. Complexity and performance comparison with the state-of-the-art methods on three public datasets.**
> | Model     | MACs  | RefCOCO Val | RefCOCO TestA | RefCOCO TestB | RefCOCO+ Val | RefCOCO+ TestA | RefCOCO+ TestB | G-Ref Val(U) | G-Ref Test(U) | G-Ref Val(G) |
> |----------------|-------|-------------|---------------|---------------|----------|----------|-----------|--------------|---------------|--------------|
> | DMMI       | 392G  | 74.13       | 77.13         | 70.16         | 63.98        | 69.73     | 57.03          | 63.46        | 64.19     | 61.98    |
> | CGFormer     | 950G  | 74.75       | 77.30     | 70.64         | 64.54        | 71.00     | 57.14          | 64.68        | 65.09         | 62.51     |
> | METRIS (Ours)  | 432G  | **75.35**  | **77.97** | **71.94**  | **66.70**  | **72.08**  | **59.85**  | **65.78** | **66.93**   | **63.49**  |
>
>
> We appreciate the reviewer's constructive comments and insightful suggestions, and we hope that our responses solve the reviewer's all concerns.

---

> ### Author Response · Authors · 2024-11-24
>
> Dear Reviewer 6fUF,
>
> There are only a few days left until the end of the Author-Reviewer Discussion phase. The reviewer's insights have been highly valued, and your feedback is crucial to our progress. We understand that you have a busy schedule. It would be really grateful if you could confirm that you have read the rebuttal and our revised version of the paper, and we would sincerely appreciate if you could adjust your evaluation accordingly. If the reviewer has any questions or concerns, we will do our best to address them quickly. Thank you for your time and effort in reviewing our paper, and we look forward to your response.

---

### Official Review · Reviewer_rFBT · 2024-10-31

**Soundness:** 2
**Presentation:** 3
**Contribution:** 2
**Rating:** 5
**Confidence:** 5

**Summary:**

This paper introduces a novel Multi-Expression guidance framework for Transformer-based Referring Image Segmentation, called METRIS, designed to improve segmentation accuracy by incorporating both visual and linguistic expressions as guidance. METRIS enhances segmentation using a visual expression extractor, which provides detailed visual context to complement linguistic guidance.

**Strengths:**

1. The proposed visual expression extractor enhances the adaptability to diverse image and language inputs and improves visual understanding of the fine-grained target regions.
2. The source code is attached, which can help the reviewers to check more model details.

**Weaknesses:**

- The SOTA comparison with [1] is missing in Table 4. Combining Tables 1 and 5 could save space, allowing for the inclusion of additional methods for comparison.
- The performance of this method does not achieve state-of-the-art compared to PolyFormer, published in CVPR 2023, as seen in Table 8.
- The contrastive loss in Equation (4) appears similar to the loss in CRIS, so the citation should be added in the appropriate place.
- The motivation for this work is unclear; it appears that visual expressions are sampled through filtering and thresholding strategies, which are then reframed as a “guidance set” from my perspective.

Other minor issues:
- In Line 225, the source of the relevance score map $s$ is not explained.
- The font size in Figures 1 and 2 is too small, causing the text to blur when zoomed in.
- Ablation: removing steps 1, 2, and 3 from the visual expression extractor could further demonstrate the effectiveness of the proposed method.

[1] GSVA: Generalized Segmentation via Multimodal Large Language Models. In CVPR 2024

**Questions:**

See weakness.

---

> ### Author Response · Authors · 2024-11-20
> **Response to Reviewer rFBT (1/n)**
>
> We sincerely thank for the insightful comment and valuable suggestions. We address the reviewer's concerns below:
>
> > **W1. SOTA comparison**
>
> - Combining Table 1 and 5 of our draft
>
> Thank you for the reviewer’s great suggestion. Reflecting the reviewer’s feedback, we combined Tables 1 and 5 of our draft in the revised paper. However, we marked the performance of Table 5’s models in grey text in Table 1 of the revised paper, because it is an unfair to directly compare the LLM-based models (i.e., the model in Table 5 of our draft) with the traditional Transformer-based models.
>
> - Comparison to GSVA [1]
>
> Thank you for recommending a great paper relevant to our study. Reflecting the reviewer’s comment, we added [1] in Table 4 of the revised paper to compare the performance on gRefCOCO. As shown in Table J, our model showed higher performance than the LLM-based model [1] on gRefCOCO benchmarks. Since gRefCOCO consists of a wide range of samples that cover more diverse target regions, these results demonstrated that our model has better ability to understand the visual contexts of the target regions than the LLM-based model.
>
> >**Table J. Comparison with GSVA [1] on the generalized referring image segmentation benchmark (gRefCOCO)**
> | Method    | val mIoU | val oIoU | testA mIoU | testA oIoU | testB mIoU | testB oIoU |
> |-----------|----------|----------|------------|------------|------------|------------|
> | GSVA-7B [1]  | 66.47    | 63.29    | 71.08      | 69.93      | 62.23      | 60.47      |
> | METRIS (Ours)    | **69.37** | **65.88** | **72.81**  | **71.74**  | **64.29**  | **63.30**  |
>
> ---
> > **W2. Performance comparison to PolyFormer**
>
> (Table 8 in our draft &rightarrow; Table 7 in the revised paper)
>
> Thanks for the reviewer's comment on the need to clarify the comparison with PolyFormer. As the reviewer mentioned, our model does not surpass Polyformer on some data splits in Table 7 of the revised paper. However, it is unfair to directly compare our model with Polyformer, because of these reasons:
> 1) Polyformer uses large-scale vision-language grounding datasets, including Visual Genome and Flickr30k-entities, to train the model for pre-training.
> 2) PolyFormer is designed for multi-task training. Training for multi-task brings the significant performance improvement of referring segmentation, as evidenced in Table 5 of PolyFormer, where the multi-task training resulted in a 3.07% gain.
>
> Our paper focuses on the referring segmentation task and does not incorporate the large-scale referring grounding datasets at training. Therefore, a direct comparison between our model and PolyFormer is unfair. Despite the unfair comparison, our model outperformed PolyFormer on several RIS datasets (5 dataset splits) in Table 7 of the revised paper.
>
> We reflected these explanations in the revised paper and we hope that our response helps address the reviewer's concerns.
>
> ---
> >**W3. The citation for the contrastive loss**
>
> Thank you for thoughtful comment. Reflecting the reviewer’s feedback, we added a citation [CRIS] at the explanation part of the contrastive loss in Method Section (line 239 of the revised paper).

---

> ### Author Response · Authors · 2024-11-20
> **Response to Reviewer rFBT (2/n)**
>
> >**W4. Motivation Clarification**
>
> Thanks for the reviewer's constructive comment, which is valuable in clarifying our motivation.
>
> In this paper, we define the guidance set as a set of key-value tokens of the cross-attention in a segmentation decoder. This is a revisiting of the part used in the existing RIS framework. Additionally, we analyzed RIS frameworks based on how the guidance set is utilized, as shown in Figure 1 of our draft, and found that previous methods rely on linguistic-based tokens (i.e., the pure linguistic expression tokens or the enhanced linguistic expression tokens) as a guidance set. Based on the understanding that the given text inherently contains the target information for guiding to the target regions, we revisit the *role* of the guidance set, extending it from a simple set of the text tokens to a set of tokens equipped with target-informative guidance ability. Furthermore, the Table 5 of the revised paper shows that the composition of the guidance set has a significant impact on RIS performance improvements.
>
> Therefore, **the motivation of our paper is not only merely the definition of the guidance set itself but rather an approach enhancing the capacity of the guidance set via the introduction of the additional target guidance information.** This is because the given language expression has the limitation due to the lack of target guidance capability, as shown in Figure 2 (a) of the revised paper.
>
> To address this limitation, *we propose a novel approach that incorporates the visual expression generated by the filtering strategies into the guidance set, deviating from the previous approach that only language-based tokens can fulfill the role of providing the target information. Our approach is the first to explore the potential of the visual expression as a provider of target
> guidance information (i.e., the guidance set) in Transformer-based referring image segmentation.* This approach shows the significant performance gains in Table 5 of the revised paper, demonstrating the importance of target-informative visual guidance ability in the guidance set. Additionally, in order to clearly clarify the effectiveness of our approach, we added more visualized comparisons between the visual expression and the language expression in Figures 7 and 11-15 of the revised paper.
>
> ---
> >**Minor Issues**
>
> - The source of the relevance score map
>
> The relevance score map $s$ indicates the relevance score map between the all vision tokens and a global linguistic token. Thus, $s$ is taken from $S_c$ of Eq. (3).
>
> - The font size in Figures 1 and 2 is too small, causing the text to blur when zoomed in.
>
> We sincerely apologize for the too small font size and the blurred images when zoomed in. We updated the Figure 1 and 2 in the revised paper to provide more clear images.
>
> - Ablation on removing all steps from the visual expression extractor
>
> Thank you for the reviewer’s insightful feedback. Reflecting the feedback, we added the ablation results for removing steps 1,2, and 3 from the visual expression extractor in Table 6 of the revised paper.
> As shown in Table K, the removal of all steps resulted in significant drops by 2.85% and 2.59% in mIoU and oIoU on average. These results demonstrated that our visual expression extractor can effectively generate the visual expression equipped with the target-informative visual guidance capability for the robust performance.
>
> >**Table K. Ablation study on removing all steps of the visual expression extractor**
> | Step 1 | Step 2 | Step 3 | RefCOCO val mIoU / oIoU| RefCOCO+ val mIoU / oIoU| G-Ref val(U) mIoU / oIoU |
> |--------|--------|--------|------|---------------|---------------|
> |✗      | ✗      | ✗      |75.10 / 73.56 | 65.46 / 63.97 | 64.35 / 63.68 |
> | ✔      | ✔      | ✔      | **76.97** / **75.35**   | **68.63** / **66.70**   | **67.85** / **66.93**   |
>
> We appreciate the reviewer's valuable comments, and we hope that our responses address the reviewer’s all concerns.

---

> ### Author Response · Authors · 2024-11-24
>
> Dear Reviewer rFBT,
>
> There are only a few days left until the end of the Author-Reviewer Discussion phase. The reviewer's insights have been highly valued, and your feedback is crucial to our progress. We understand that you have a busy schedule. It would be really grateful if you could confirm that you have read the rebuttal and our revised version of the paper, and we would sincerely appreciate if you could adjust your evaluation accordingly. If the reviewer has any questions or concerns, we will do our best to address them quickly. Thank you for your time and effort in reviewing our paper, and we look forward to your response.

---

### Official Review · Reviewer_9BXs · 2024-11-03

**Soundness:** 2
**Presentation:** 3
**Contribution:** 1
**Rating:** 3
**Confidence:** 5

**Summary:**

This submission focuses on the problem of Transformer-based referring image segmentation (RIS) and introduces the idea that Transformer-based RIS methods utilize guidance information to guide the prediction of referring regions. However, the submission proposes that existing guidance approaches rely solely on linguistic tokens. To address this limitation, they propose a visual expression extractor that generates visual expression tokens as a complement. Experiments are conducted on three public RIS benchmarks.

**Strengths:**

- The submission is well-written and easy to follow. Additionally, the figures and tables are well-prepared and clearly illustrate the motivations and methods.
- The idea of using visual tokens to complement guidance is straightforward.
- Experiments are conducted on multiple common benchmarks and across different settings.

**Weaknesses:**

- From both the ablation study and comparisons (noting that some SOTAs are not included), the performance improvement of the proposed METRIS is limited and does not effectively demonstrate its effectiveness.
    - In Table 1, the improvement of METRIS over the previously best-performing methods is quite limited:
        - mIoU metric. The submission does not report the results for CGFormer, but the performance of this submission is only comparable to that of CGFormer. And the comparison between CGFormer and the submission is listed below:

            CGFormer  76.93 78.70 73.32 68.56 73.76 61.72 67.57 67.83 65.79

            METRIS       76.97 78.89 73.63 68.63 73.88 61.94 67.85 67.97 65.86

        - oIoU metric. The improvement of the proposed METRIS relative to LQMFormer is minimal.
    - In Table 1, some of the latest SOTAs, such as those[1][2] published in CVPR 2024 for RIS, are not included or compared. The performance of METRIS is lower than that of these methods.

        [1] Prompt-Driven Referring Image Segmentation with Instance Contrasting

        [2] Mask Grounding for Referring Image Segmentation

    - In Table 6, the baseline itself achieves good results, and the improvement over the baseline is quite limited.
- The novelty and technical contribution are insufficient.
    - Previous methods, even when utilizing "linguistic tokens," incorporate visual-language alignment and querying modules that effectively embed visual information within those tokens. They are referred to as "linguistic tokens," but they effectively function like visual tokens due to the visual-language alignment and querying modules. This paper merely makes the use of visual tokens explicit to provide further supplementation, which is an incremental gain.
    - Essentially, this approach adds attention between linguistic and visual tokens to aggregate visual information, which is a common practice in the vision-language domain.
- Table 6 does not include results for RefCOCO+ validation, while Table 7 presents results for the visual expression extractor only for RefCOCO+ validation. Both tables need to provide complete results to better understand the role of the visual expression extractor within the overall framework and the contribution of its individual modules.
- The qualitative results provide a comparison; however, they do not sufficiently demonstrate that the improvements are attributable to the visual tokens. A more in-depth comparison and analysis are required.
- Minor (does not affect rating): The figures appear blurry. Please use vector graphics or high-resolution images.

**Questions:**

- Visual tokens may encompass multiple regions, some of which could be referring regions and others that are not. Could the non-referring regions potentially act as distractions?

---

> ### Author Response · Authors · 2024-11-20
> **Response to Reviewer 9BXs (1/n)**
>
> Thank you for the reviewer’s valuable comments. We address the reviewer's concerns below:
> >**W1. The performance improvement of our METRIS**
> - Comparison with CGFormer
>
> As the reviewer mentioned, our mIoU performance is only comparable to that of CGFormer. However, our model showed comparable mIoU performance with 45.5% less computations than CGFormer, as shown in Table D. Furthermore, our model showed remarkable improvements of 1.38% oIoU on average.
>
> >**Table D. Performance comparison to CGFormer on three public RIS benchmarks**
> | Metric  | Model  | MACs  | RefCOCO Val | RefCOCO TestA | RefCOCO TestB | RefCOCO+ Val | RefCOCO+ TestA | RefCOCO+ TestB | G-Ref Val(U) | G-Ref Test(U) | G-Ref Val(G) |
> |---|-------|------|-----|-------|-----|------|-----|----|----|-----|---|
> | mIoU  | CGFormer| 950G  | 76.93       | 78.70  | 73.32  | 68.56     | 73.76 | 61.72   | 67.57   | 67.83  | 65.79  |
> |mIoU     | METRIS (Ours)  | **432G**  | **76.97** | **78.89** | **73.63**  | **68.63**   | **73.88**    | **61.94**   | **67.85**  | **67.97**         | **65.86**  |
> | oIoU   | CGFormer  | 950G  | 74.75       | 77.30  | 70.64 | 64.54  | 71.00   | 57.14   | 64.68    | 65.09  | 62.51  |
> |   oIoU    | METRIS (Ours)  | **432G**  | **75.35** | **77.97**   | **71.94**   | **66.70**   | **72.08**    | **59.85**   | **65.78**  | **66.93**  | **63.49** |
>
> ----------------
> - Comparison with LQMFormer
>
> LQMFormer has a smaller performance drop than our model only on a certain split (i.e., RefCOCO+ testA), but it performs worse than our model on all datasets, as shown in Table E. In particular, compared to our model, LQMFormer had much lower performance by 0.79% and 2.26% on the other two splits of RefCOCO+. These results indicate that LQMFormer may overfit to the certain data samples because the target object category of the testA subset is mostly a person. In contrast, our model consistently has strong performance on all splits.
>
> >**Table E. Performance comparison to LQMFormer on three public RIS benchmarks using oIoU**
> | Model   | RefCOCO Val | RefCOCO TestA | RefCOCO TestB | RefCOCO+ Val | RefCOCO+ TestA | RefCOCO+ TestB | G-Ref Val(U) | G-Ref Test(U) | G-Ref Val(G) |
> |----|----|----|----|----|----|----|-----|----|----|
> | LQMFormer | 74.16  | 76.82  | 71.04   | 65.91| 71.84 | 57.59 | 64.73 | 66.04 | 62.97  |
> | METRIS (Ours)  | **75.35**   | **77.97**   | **71.94**  | **66.70** | **72.08**  | **59.85**  | **65.78** | **66.93** | **63.49** |
>
> ----
> - Some of the latest SOTAs are not included in Table 1
>
> Thank you for recommending great papers relevant to our research.
>
> However, it is unfair to directly compare our model with [1], because [1] adopts a strong segmentation model SAM, which is trained with a large amount of segmentation datasets. Adopting SAM as the baseline model results in significant performance gains, as evidenced in the ablation results of [1], where their model without SAM resulted in 3.79% and 4.13%  drops in oIoU and mIoU than their model with SAM on RefCOCO val. As shown in Table F, compared to our model on RefCOCO val, their model without SAM also has 2.78% and 3.0% lower performance in oIoU and mIoU, respectively.
>
> >**Table F. Performance comparison to Prompt-RIS [1] on RefCOCO val**
> | Model  | P@0.5 | P@0.7 | P@0.9 | mIoU  | oIoU  |
> |----|----|-----|-----|-----|-----|
> | Prompt-RIS [1]  (w/o SAM)  | 85.55 | 76.90 | 26.16 | 73.97 | 72.57 |
> | METRIS (Ours)   |  **86.71**| **78.30** | **37.24** | **76.97** | **75.35** |

---

> ### Author Response · Authors · 2024-11-20
> **Response to Reviewer 9BXs (2/n)**
>
> Reflecting the reviewer’s comment, we added [2] into Table 1 of the revised paper.
>
> [2] shows better performance on RefCOCO testA in Table G. However, except for RefCOCO testA, our model showed higher performance with an average of 0.72% on 8 datasets compared to [2]. Particularly, compared to [2], our model showed significant improvements by 0.89%, 0.76%, and 1.71% in oIoU on RefCOCO testB, RefCOCO+ testA and testB, respectively. We analysed these results based on dataset characteristics:
> 1) On RefCOCO and RefCOCO+, the testA set contains images of multiple people and the testB set contains images of multiple instances of all other objects. In test A set, the target object category of 93% of the data samples is a person.
> 2) RefCOCO+, which forbids the words about the absolute locations in the language expressions, is more challenging than RefCOCO.
>
> According to the analysis, the performance comparison results indicate that the robustness of our model is stronger than that of [2] for a wider variety of object inputs and difficult language inputs, whereas [2] may overfit to the target object of people (i.e., testA) on RefCOCO. In other words, our method has better generalizability than [2] for the diverse image-language scenarios.
>
> >**Table G. Performance comparison to MagNet [2] on three public RIS benchmarks**
> | Model          | RefCOCO Val | RefCOCO TestA | RefCOCO TestB | RefCOCO+ Val | RefCOCO+ TestA | RefCOCO+ TestB | G-Ref Val(U) | G-Ref Test(U) | G-Ref Val(G) |
> |----------------|-------------|---------------|---------------|--------------|----------------|----------------|--------------|---------------|--------------|
> | MagNet         | 75.24       | **78.24**         | 71.05         | 66.16        | 71.32          | 58.14          | 65.36        | 66.03         | 63.13        |
> | METRIS (Ours)  | **75.35**       | 77.97         | **71.94**         | **66.70**        | **72.08**          | **59.85**          | **65.78**        | **66.93**         | **63.49**        |
>
> -------------------
> -  The improvement over the baseline in Table 5 of the revised paper (Table 6 of our draft)
>
> We are unsure which ablation model the reviewer referred to as baseline in Table 5 of the revised paper. Our baseline method is an ‘Enhanced LE’ method (row2) in Table 5, because previous methods have exploited the vision-aware language tokens (i.e., ‘Enhanced LE’) for performance improvements, as the reviewer mentioned in the next comment below. However, our method allows the introduction of the target-informative visual tokens as guidance elements. Therefore, compared to ‘Enhanced LE’ method, our method showed improvements of 1.79%, 2.73%, and 3.25% in oIoU on RefCOCO, RefCOCO+ and G-Ref, respectively. We believe that these are significant improvements on segmentation-based tasks. Reflecting the reviewer's comment, we clarified the baseline model in the ablation section (line 403) of the revised paper.
>
> Additionally, the comparison between Row 1 and 2 is a comparison between the existing frameworks, which leverage only language guidance, to adopt the baseline. The Rows 3,4 and 5, which leverage visual guidance, are considered to be the ablation models based on our visual guidance approach.

---

> ### Author Response · Authors · 2024-11-20
> **Response to Reviewer 9BXs (3/n)**
>
> >**W2. Novelty & technical contribution**
>
> As the reviewer mentioned, previous methods (i.e., ‘Enhanced LE’ method in our draft) use the cross-attention module to aggregate the visual information within the linguistic tokens by using the linguistic tokens as query and the vision tokens as key-value. We also agree that it is common practice to provide visual information to the segmentation network via these linguistic-based tokens.
>
> However, as shown in Figure 2, we found that linguistic expression tokens have limitations due to the lack of target information, despite exploiting the Enhanced LE tokens. Technically, our method leverages visual expression tokens by concatenating them with the enhanced linguistic expression tokens as elements of the guidance set that are used as key-value in the segmentation decoder, as shown in Figure 3. Therefore, **this is totally different from adding the attention between the linguistic and vision tokens, and can be considered as adding the attention between the vision features (as a query) and the target-informative visual tokens (i.e., visual expression) (as a key-value) in the decoder.** To resolve this confusion, we modified Figures 1 & 3 of the revised paper by clearly notating the key-value.
>
> In addition, our approach shows a novel perspective on how to effectively leverage the target information contained in the vision and language features, compared to existing methods that simply aggregate the vision and language features. Especially, *our study is the first approach to generate the visual expression, which is equipped with the target-informative visual guidance capability, as a provider of the target information, deviating from previous approaches in that only linguistic-based tokens (e.g., Enhanced LE tokens) can fulfill the role of providing target information to the network. In other words, our approach is the first to explore the potential of the visual expression as a provider of target guidance information in Transformer-based RIS.*
>
> Quantitatively, compared to the ablation model using only the Enhanced LE, our method resulted in improvements of 2.59% and 2.85% in oIoU and mIoU on average in Table 5 of the revised paper. Qualitatively, we added visual comparisons in Figure 7 of the revised paper, which more clearly clarify the effectiveness of our method. Our VE is particularly effective in correcting the incorrect regions, whereas the Enhanced LE includes wide non-target regions or misses target regions. We additionally displayed more qualitative comparisons in Figures 11-14 of the revised paper.
>
> ---
> >**W3. Ablation results for RefCOCO+ val in Table 6 of our draft & for RefCOCO val and G-Ref val in Table 7 of our draft**
>
> (Table 6 &7 in our draft &rightarrow; Table 5 & 6 in the revised paper)
>
> We had adopted different datasets in Tables 6 and 7 to show the results of ablation studies on different datasets in our draft, but we agree that both tables need to provide complete results to better understand the role of the visual expression extractor.
>
> Reflecting the constructive comment, we added ablation results on RefCOCO+ val into Table 5 of the revised paper, and on RefCOCO val and G-Ref val into Table 6 of the revised paper. For a fair comparison, we added the cross-attention layers into the ablation models to maintain the model size similar to our default model. As shown in Table H, our full model showed performance gains of 3.17% and 2.73% in mIoU and oIoU on RefCOCO+ val, compared to the ablation model using only the Enhnaced LE (baseline model). This result indicates that our visual expression approach is effective in enhancing the capacity of the guidance set for robust performance. In Table I, the removal of each component resulted in performance drops by 0.84% and 1.96% oIoU on average on RefCOCO val and G-Ref val. This result demonstrates that each component is necessary to generate the visual expression equipped with the target-informative visual guidance capability.
>
> >**Table H. Ablation for effectiveness of our guidance set on RefCOCO+ val.**
> LE : Linguistic Expression, VE: Visual Expression (Ours)
> |Linguistic|Visual|P@0.5|P@0.7|P@0.9|mIoU|oIoU|
> |---|---|---|---|---|---|---|
> |Pure LE|✗|73.54|64.59|28.35|63.72|62.15|
> |Enhanced LE|✗|74.90|66.12|29.83|65.46|63.97|
> |✗|VE|76.29|67.60|31.36|67.33|65.59|
> |Enhanced LE|All pixels|75.81|67.28|30.89|66.97|65.24|
> |Enhanced LE|VE| **77.13** | **69.05** | **32.94**| **68.63** | **66.70** |
>
> >**Table I. Ablation on design of our visual expression extractor**
> |Step1|Step2|Step3|RefCOCO val mIoU/oIoU|G-Ref val mIoU/oIoU|
> |---|---|---|---|---|
> |✗|✗|✗|75.10/73.56|64.35/63.68|
> |✗|✔|✔|76.09/74.50|66.01/65.13|
> |✔| ✗|✔|75.98/74.44|65.79/64.77|
> |✔|✔|✗|76.13/74.63|66.26/65.25|
> |✔|✔|✔| **76.97** / **75.35** | **67.85** / **66.93** |
>
> >|Global|Local|RefCOCO val mIoU/oIoU|G-Ref val mIoU/oIoU|
> |---|---|---|---|
> |✔|✗|76.20/74.43|65.83/64.71|
> |✗|✔|76.26/74.55|66.65/64.99|
> |✔|✔| **76.97** / **75.35** | **67.85** / **66.93** |

---

> ### Author Response · Authors · 2024-11-20
> **Response to Reviewer 9BXs (4/n)**
>
> >**W4. Qualitative results for more in-depth comparison and analysis**
>
> Thank you for the reviewer’s insightful comment on the in-depth analysis of our method. To address this, we examined the roles of the enhanced language expression and the visual expression in our model, which leverages both expressions, through the detailed analysis of attention maps in Figures 7(b) and 15 of our revised paper.
>
> In the segmentation decoder using our guidance set, the cross-attention uses the vision features (i.e., all vision tokens) as a query and the guidance set, which consists of both the enhanced language expression and the visual expression (i.e., target-informative visual tokens), as a key-value. Therefore, we compared the attention map between the vision features and the enhanced language expression with the attention map between the vision features and the visual expression.
>
> Surprisingly, the results show that **while the enhanced language expression tends to miss large part of the target regions or attend to even wide non-target regions, the visual expression demonstrates strong alignments with the target regions of the vision features.** These results indicate that the proposed visual expression effectively enhances the capability to align with the target regions, especially where the enhanced linguistic expression is insufficient. We hope that this experiment helps to clarify our approach and provides further insights.
>
> --------
> >**W5. Minor issue for the blurred figures**
>
> We sincerely apologize for the blurred images. We updated the Figures in the revised paper to provide more clear images.
>
> ------------
> >**Q1. Visual tokens may encompass multiple regions, some of which could be referring regions and others that are not. Could the non-referring regions potentially act as distractions?**
>
> As the reviewer’s question, we also have considered the possibility that non-target information, which can act as distraction, may be included in the visual expression during the process of generating the visual expression, because the image features contain both target and non-target information.
>
> To address this, we conducted various experiments to explore the design choices for implementing our method effectively. First, we leverage the token curation based on the cosine similarity function to obtain visual tokens that are closely embedded to the linguistic expression tokens in the multi-modal embedding space. For the token curation, as shown in Figure 6 (a), we conducted an ablation study to determine the optimal filtering ratio (i.e., the number of tokens curated from total tokens). The results showed that the optimal ratio was approximately 30% of the total tokens. The larger number of the ratio resulted in the performance degradation due to noise information that can interfere with the target information, and the smaller ratio also decreased the performance due to the lack of target information. Furthermore, as shown in Table 6 (a) of the revised paper, the removal of the curation process (Step1) resulted in a performance drop of 1.55% oIoU on average, demonstrating the effectiveness of selectively leveraging the target-informative visual tokens among the total visual tokens that include both target and non-target information.
>
> Despite performing the token curation in our framework, non-referring region tokens may be encompassed in the set of the curated tokens. To prevent the information from being distracted by these tokens, the Step 2 of our module was designed to adaptively refine the semantic information from the curated tokens, rather than simply aggregating the curated tokens. As shown in Figure 3, our refinement step creates the representative visual tokens by aggregating for each curated token set, and then these representative visual tokens are used to compute attention with the curated tokens of the corresponding set, adaptively applying the attention weights to relevant and irrelevant tokens. This process effectively reduces distracting information by adaptively focusing on the semantic tokens. Furthermore, as shown in Table 6 (a) of the revised paper, the removal of this refinement process (Step2) resulted in a performance drop of 1.69% on average. This demonstrates that adaptively capturing the semantic information from the curated tokens is more effective at preventing the distracting information than simply aggregating the curated tokens.
>
> We sincerely appreciate the reviewer's constructive and insightful comments, which are valuable in clarifying our contributions. We hope that our responses address all of the reviewer's concerns.

---

> ### Author Response · Authors · 2024-11-24
>
> Dear Reviewer 9BXs,
>
> There are only a few days left until the end of the Author-Reviewer Discussion phase. The reviewer's insights have been highly valued, and your feedback is crucial to our progress. We understand that you have a busy schedule. It would be really grateful if you could confirm that you have read the rebuttal and our revised version of the paper, and we would sincerely appreciate if you could adjust your evaluation accordingly. If the reviewer has any questions or concerns, we will do our best to address them quickly. Thank you for your time and effort in reviewing our paper, and we look forward to your response.

---

> > ### Comment · Reviewer_9BXs · 2024-11-27
> > **Response to Author Rebuttal**
> >
> > Thank you to the author for the thorough and detailed response, which has addressed some of my concerns. However, I remain concerned about the limited improvements and the incremental technical contribution:
> > * The improvements over SoTA methods across different splits are almost all below 1%. This makes it difficult to sufficiently validate the effectiveness of the method.
> > * I understand that this paper concatenates the vision expression tokens with the linguistic tokens as a guidance set, rather than using cross-attention between them. However, this is a highly incremental modification at the module level and does not constitute a significant technical contribution.

---

> > > ### Author Response · Authors · 2024-11-27
> > >
> > > We genuinely appreciate the reviewer's response and constructive feedback for our work. We additionally addressed the reviewer's concerns below:
> > >
> > > > **1. Improvements over SOTA methods**
> > >
> > > - When comparing the performance between the more recent SOTA models (e.g., MagNet, LQMFormer) and the earlier SOTA models (e.g., CGFormer, DMMI, ReLA) in Table X, more recent SOTA models' improvements over the earlier SOTA models are all below 1%. MagNet shows 0.06%~0.94% improvements over the earlier SOTA models, and LQMFormer shows performance improvements on only four data splits. This comparison suggests that it is challenging to improve performance on all datasets, and that the performance improvements of less than 1% over the SOTA models are not marginal improvements in this task.
> > >
> > > >**Table X. Performance comparison with the recent SOTA methods on three RIS benchmarks**
> > > | Method| RefCOCO val | RefCOCO testA | RefCOCO testB | RefCOCO+ val | RefCOCO+ testA | RefCOCO+ testB | G-Ref val (U) | G-Ref test (U) | G-Ref val (G) |
> > > |-----|------|------|------|---|-----|------|------|-----|-----|
> > > | ReLA (CVPR2023)  | 73.82  | 76.48   | 70.18     | 66.04 | 71.02      | 57.65    | 65.00   | 65.97      | 62.70   |
> > > | DMMI (ICCV2023)  | 74.13| 77.13 | 70.16  | 63.98   | 69.73   | 57.03   | 63.46     | 64.19   | 61.98   |
> > > | LQMFormer (CVPR2024) | 74.16    | 76.82   | 71.04   | 65.91    | 71.84     | 57.59      | 64.73     | 66.04    | 62.97 |
> > > | CGFormer (CVPR2023)  | 74.75    | 77.30     | 70.64    | 64.54  | 71.00     | 57.14     | 64.68     | 65.09     |62.51    |
> > > | MagNet (CVPR2024) | 75.24  |  **78.24**  | 71.05    | 66.16    | 71.32     | 58.14     | 65.36    | 66.03    |63.13 |
> > > | METRIS (Ours) | **75.35**   | 77.97    | **71.94**  | **66.70**  | **72.08**   | **59.85**      | **65.78**   | **66.93** | **63.49**  |
> > >
> > >
> > > - In Table Y, we also additionally experimented on applying our visual expression approach to MagNet as the baseline, which leverages the auxiliary task loss, to validate the effectiveness of our approach. Our method brings significant gains on each dataset. This result indicates that using the visual expression can be universally applied to other models, and demonstrates the effectiveness of our approach.
> > >
> > > > **Table Y. Applying our visual expression to the recent SOTA model**
> > > | Method          | RefCOCO val | RefCOCO testA | RefCOCO testB | RefCOCO+ val | RefCOCO+ testA | RefCOCO+ testB | G-Ref val (U) | G-Ref test (U) | G-Ref val (G) |
> > > |--------|--------|-------|-------|-------|-------|------|------|------|------|
> > > | MagNet   | 75.24   | 78.24   | 71.05   | 66.16   | 71.32   | 58.14    | 65.36  | 66.03     | 63.13   |
> > > | MagNet + Ours   | **76.73**  | **79.56**    | **72.58**   | **67.77**  | **72.97**  | **60.18**   | **67.06**    | **67.30** | **63.92**  |
> > >
> > > ---
> > > > **2. Technical contribution**
> > >
> > > We understand that using the concatenation of the visual expression tokens and the linguistic expression tokens may seem to be an incremental modification. However, **our technical contribution is to generate the visual expression tokens equipped with the target-informative visual information from the original vision tokens, because the original vision tokens contain distracting information as mentioned in the reviewer's question.**
> > >
> > > Our method is the first to generate the visual expression via the retrieval of the target-informative visual information, which overcomes the lack of guidance capability provided by the linguistic expression. We believe that this provides a new perspective on the vision-language task and could bring an insight for a variety of fields.
> > >
> > > Additionally, in order to more rigorously compare the performance gain with the structure mentioned by the reviewer 9BXs, we conducted an experiment on adding the cross-attention modules between the linguistic and vision tokens instead of the visual expression extractor. As shown in Table Z, using visual expression tokens brings remarkable improvements of 1.33%, 1.89% and 2.39% on each dataset. These results highlight that our modification is highly effective for referring image segmentation compared to the cross-attention between the linguistic and vision tokens.
> > >
> > > We are truly grateful for the reviewer's feedback, which allowed us to conduct meaningful additional experiments. We sincerely hope that our response addresses the reviewer's all concerns.
> > >
> > > >**Table Z. Ablation on adding cross-attention modules between linguistic and vision tokens instead of the visual expression extractor**
> > > | Method | RefCOCO val (mIoU/oIoU) | RefCOCO+ val (mIoU/oIoU) | G-Ref val (U) (mIoU/oIoU) |
> > > |----|----|------|------|
> > > |Enhanced LE + adding the cross-attention modules between linguistic and vision tokens    | 75.48 / 74.02 | 66.28 / 64.81   | 65.30 / 64.54 |
> > > |Enhanced LE + using visual expression (Ours)    | **76.97** / **75.35**   | **68.63** / **66.70**  | **67.85** / **66.93**  |

---

### Official Review · Reviewer_xfB1 · 2024-11-06

**Soundness:** 3
**Presentation:** 3
**Contribution:** 2
**Rating:** 5
**Confidence:** 4

**Summary:**

This paper propose a new model for referring image segmentation. The authors propose a visual expression extractor to selectively model the shared representation between visual and language modality. The proposed model achieve new state-of-the-art on various referring image segmentation benchmarks.

**Strengths:**

+ The paper is overall easy to understand

+ The proposed model achieve decent performance on various referring image segmentation benchmarks

+ The authors provide various analysis such as generalization and various ablation studies.

**Weaknesses:**

- The contribution is limited. The authors argue that previous methods cannot capture the target-informative visual understanding, but I believe they can do via cross-attentions. And actually, the proposed method is a set of cross-attentions. I think the proposed method consists of several cross-attentions and token selection that is popularly used in the recent literatures in various fields.

- The ablation studies are not through. To show the effectiveness of the proposed component, I recommend the authors to add more cross-attention mechanism to keep the number of total learnable parameters similar.

- Comparison with recent SOTA methods is missing. For example, [ref1] produces better results on RefCOCO test A. [ref2] produces better results on many benchmarks. And, [ref3] is also worth to be mentioned in the paper

[ref1] Mask Grounding for Referring Image Segmentation, CVPR 2024
[ref2] Prompt-Driven Referring Image Segmentation with Instance Contrasting, CVPR 2024
[ref3] SAM4MLLM: Enhance Multi-Modal Large Language Model for Referring Expression Segmentation, ECCV 2024

**Questions:**

1. I'm curious why the proposed method is strong on unseen classes. the proposed method learns to attend/correlate for seen categories, so may be overfitted to seen classes and may not be generalized to unseen categories. More motivations and analysis on unseen classes would be great.

---

> ### Author Response · Authors · 2024-11-20
> **Response to Reviewer xfB1 (1/n)**
>
> Thank you for the reviewer’s constructive comments. We address the reviewer's concerns below:
>
> >**W1. Contributions of our method compared with the cross-attention based previous method (i.e. Enhanced linguistic token)**
>
> We sincerely appreciate the reviewer’s insightful comments regarding the contribution of our work.
>
> As the reviewer mentioned, we agree that most previous methods leverage the visual-aware linguistic tokens, which capture visual understanding into linguistic tokens via the cross-attention. This can improve RIS performance by enhancing the capability of the vision-language alignment (i.e., the alignment between the target regions and the target-informative elements). We also use the visual-aware linguistic tokens (called as the enhanced linguistic expression tokens in our draft) as one of the guidance elements.
>
> However, **our key contribution** is to exploit not only the “vision-language” alignment, but also the “vision-target informative vision”  alignment for guiding the network to the target region (i.e., 'target informative vision' indicates visual expression). By additionally introducing our “vision-target informative vision” alignment process, our method significantly complements the target information that is not captured in the enhanced linguistic expression. In particular, *our method has a **novelty** in that it produces the visual expression, which is equipped with the target-informative visual guidance capability, as a provider of the target information, beyond the previous approaches in that only language-based tokens (e.g., the visual-aware linguistic tokens) can fulfill the role of providing the target information to the network.* In other words, *our approach is the first to explore the potential of the visual expression as a provider of target guidance information in Transformer-based referring image segmentation.*
>
> In addition, the effectiveness of our approach is demonstrated in the ablation Table 5 of the revised paper. To provide in-depth analysis of our method and further clarify the effectiveness of our approach, we additionally displayed the attention maps of the enhanced linguistic expression and the proposed visual expression in Figures 7 and 15 of the revised paper. The results showed that our visual expression remarkably complements the target information even though the enhanced language expression misses the target regions or includes even non-target regions, addressing the misalignments caused by the visual-aware linguistic token’s limitation.
>
> Additionally, as the reviewer mentioned, the cross-attention and the token selection are commonly-used techniques in various fields. However, we believe that the importance lies *not* in the technique itself, *but* in how its structure and its methodology are purposefully applied to achieve specific objectives. In other words, the most critical aspect in the cross-attention is how the query, key and value are utilized, as the way they are defined can lead to entirely different roles, even within the same structure. Especially, our method is the first to enhance the target guidance capacity by leveraging the *visual expression tokens*, which are endowed with the target-informative guidance ability via the token selection, as the key-value (i.e., the guidance set) in the cross-attention, beyond leveraging the enhanced linguistic expression tokens as key-value.
>
> Furthermore, the ablation comparison in Table 5 and Figures 7 & 11-14 demonstrated that the guidance element using as key-value in the cross-attention mechanism has a significant impact on the performance improvements in this task, and the ablation study in Table 6(a) of our paper demonstrated that the token curation step is necessary to endow with the target-informative guidance capability.
>
> We hope that this response addresses the reviewer’s concerns.
>
> ---
> >**W2. Ablation study to show the effectiveness of our proposed component with similar number of learnable parameters**
>
> We apologize for the missing information about the ablation model settings. As the reviewer mentioned, for a fair comparison in all ablation experiments of our draft, we have set the parameters of the ablation models to be consistent with the parameters as a 278M of our full model by adding cross-attention layers.
>
> Under the similar parameter setting, our full model significantly outperformed the ablation models in Table 5 and Table 6 of the revised paper, demonstrating the effectiveness of the proposed method. We added the explanation for the ablation model setting into the ablation study section (line 377 and 426) of the revised paper. We hope that this clarification addresses the reviewer's concerns.

---

> ### Author Response · Authors · 2024-11-20
> **Response to Reviewer xfB1 (2/n)**
>
> > **W3. Comparison with recent SOTA methods**
>
> Thank you for recommending great papers relevant to our research.
>
> - Comparison with [ref1]
>
> Reflecting the reviewer’s feedback, we added [ref1] in Table 1 of the revised paper. As the reviewer mentioned, except for RefCOCO testA, our model showed higher performance on the remaining 8 datasets compared to [ref1], as shown in Table A. Especially, compared to [ref1], our model showed significant improvements by 0.89%, 0.76%, and 1.71% in oIoU on RefCOCO testB, RefCOCO+ testA and testB, respectively. We analysed these comparison results based on dataset characteristics:
> 1) On RefCOCO and RefCOCO+, the testA set contains images of multiple people and the testB set contains images of multiple instances of all other objects. In test A set, the target object category of 93% of the data samples is a person.
> 2) RefCOCO+, which forbids the words about the absolute locations in the language expressions, is more challenging than RefCOCO.
>
> Therefore, the comparison results indicate that our method shows more robust predictions than [ref1] for a wider variety of object inputs and difficult language inputs, whereas [ref1] may overfit to the target object of people (i.e., testA set) on RefCOCO. In other words, our method has better generalizability than [ref1] for the diverse image-language scenarios.
>
> >**Table A. Peformance comparison with MagNet [ref1] on three public RIS datasets**
> | Model          | RefCOCO Val | RefCOCO TestA | RefCOCO TestB | RefCOCO+ Val | RefCOCO+ TestA | RefCOCO+ TestB | G-Ref Val(U) | G-Ref Test(U) | G-Ref Val(G) |
> |----|----|----|-----|----|-----|----|------|----|----|
> | MagNet [ref1] | 75.24   | **78.24**| 71.05   | 66.16    | 71.32 | 58.14    | 65.36   | 66.03    | 63.13  |
> | METRIS (Ours)  | **75.35**  | 77.97    | **71.94**    | **66.70** | **72.08** | **59.85**    | **65.78**   | **66.93**    | **63.49**|
>
> ---
> - Comparison with [ref2] and [ref3]
>
> It is unfair to directly compare our model with [ref2] and [ref3], because of these reasons :
> 1) [ref2] adopts a strong segmentation model SAM, which results in considerable performance improvements in RIS task because of being trained with a large amount of segmentation datasets, as evidenced in the ablation results of [ref2], where their model without SAM resulted in 3.79% and 4.13% drops in oIoU and mIoU than their model with SAM on RefCOCO val. Compared to our model on RefCOCO val, as shown in Table B, their model without SAM also showed 2.78% and 3.0% lower performance in oIoU and mIoU, respectively.
> 2) [ref3] is a LLM-based model, and adopts a strong segmentation model SAM. However, as the reviewer mentioned, it is worth to be mentioned as LLM-based RIS model. Therefore, we added it to the LLM-based model section of the Table 1 in the revised paper.
>
> >**Table B. Comparison with Prompt-RIS [ref2] on RefCOCO val**
> | Model   | P@0.5 | P@0.7 | P@0.9 | mIoU  | oIoU  |
> |----|-----|----|-----|----|----|
> | Prompt-RIS [ref2] (w/o SAM) | 85.55 | 76.90 | 26.16 | 73.97 | 72.57 |
> | METRIS (Ours)  |  **86.71**| **78.30** | **37.24** | **76.97** | **75.35** |
>
> ---
> >**Q1. Analysis on unseen classes**
>
> We appreciate the reviewer recognizing that our method is strong on unseen classes, and thank you for the reviewer’s insightful question, which offers a new perspective. Referring image segmentation task fundamentally use a prompt to segment the target region. Moreover, many previous studies [1, 2] on the multi-modal prompt learning for the zero-shot task have shown that using the dynamic prompt conditioned on vision data contributes to improve generalization.
>
> Our method leverages not only the given language inputs, but also the corresponding visual expression as a target prompt. In addition, the visual expression is dynamically generated depending on the image context (vision input condition), even when the same language expression is given as a language input, thus enhancing the model’s generalization capability.
>
> To verify this, we compared with the ablation model (i.e., ‘Enhanced LE’ model) using only the enhanced language expression on unseen setting in Table C. For a fair comparison, we maintained the model size similar to our default model. Compared to our method, the ablation model showed performance drops of 2.56%, 1.17%, and 3.71% on each dataset, respectively.
>
> >**Table C. Ablation study on unseen classes.**
> LE : Linguistic Expression , VE : Visual Expression (Ours)
> |Linguistic|Visual| val (U)|test (U)|val(G)|
> |---|---|---|---|---|
> |Enhanced LE|✗| 44.18 | 41.89 | 42.30 |
> |Enhanced LE|VE| **46.74** | **43.06** | **46.01** |
>
> [1] Zhou, Kaiyang, et al. "Conditional prompt learning for vision-language models." CVPR, 2022.
> [2] Khattak, Muhammad Uzair, et al. "Maple: Multi-modal prompt learning." CVPR, 2023.
>
> We sincerely appreciate the reviewer's insightful comments, which are valuable in clarifying our contribution. We hope that our explanations address the reviewer's all concerns.

---

> ### Author Response · Authors · 2024-11-24
>
> Dear Reviewer xfB1,
>
> There are only a few days left until the end of the Author-Reviewer Discussion phase. The reviewer's insights have been highly valued, and your feedback is crucial to our progress. We understand that you have a busy schedule. It would be really grateful if you could confirm that you have read the rebuttal and our revised version of the paper, and we would sincerely appreciate if you could adjust your evaluation accordingly. If the reviewer has any questions or concerns, we will do our best to address them quickly. Thank you for your time and effort in reviewing our paper, and we look forward to your response.

---

### Author Response · Authors · 2024-11-20

We sincerely appreciate the constructive comments and insightful suggestions from all of the reviewers. We value the feedback from our reviewers and have carefully addressed their concerns in our revised paper in response to their comments.

We have uploaded a revision of our paper where new material is highlighted in **blue** to aid reviewers.
Several Table numbers are changed in the revised paper : Table **6**, **7**, and **8** in our draft &rightarrow; Table **5**, **6**, and **7** in the revised paper

In the revised paper, we clarified sentences and provided additional experiments and qualitative results, which can be summarized as:
- **Clarifying the motivation and contribution in the introduction section** by refining explanations and modifying Figures 1 & 3.
-	Adding ablation results on RefCOCO+ val in Table 5 & on RefCOCO val and G-Ref val in Table 6 to **provide complete results on all three benchmarks for better understanding the role of the visual expression extractor**.
-	**Adding ablation results** on removing all three steps of our visual expression extractor in Table 6 (a)
-	Providing the additional **qualitative comparison of the predictions** in Figures 7(a) and 11-14 to **further clarify the effectiveness of the visual expression on diverse target region samples**.
-	Providing the **visual analysis of the attention maps** in Figures 7(b) and 15 to demonstrate that **the visual expression accurately capture the semantic target regions where the visual-aware linguistic expression fails to capture.**
-	Providing the **additional ablation** on the use of the article tokens (i.e., “the”, “a” and “an”) in Table 9

---

### Meta-Review · Area_Chair_KfEv · 2024-12-17

**Metareview:**

The paper proposes an intriguing approach, namely METRIS, for Transformer-based Referring Image Segmentation. It does so by introducing a multi-expression guidance framework that fuses visual and linguistic expressions. The concept of utilizing visual expressions to overcome the limitations of relying merely on linguistic tokens for guidance is indeed promising and holds the potential to enhance performance and visual comprehension in RIS tasks.
During the review process, the primary concerns revolve around (a) the motivation and contribution, and (b) the relatively modest improvements compared to state-of-the-art methods. The authors have endeavored to address these questions during the rebuttal. The Area Chair has carefully examined the paper and all the reviews and concurs that this paper requires further refinement prior to submission to the next iteration.

**Additional Comments On Reviewer Discussion:**

The principal concerns are as follows: (a) the contribution is deemed limited, (b) there are insufficient ablation studies, and (c) the state-of-the-art (SoTA) baselines.

During the rebuttal, in regard to (a), the authors contend that this study is the first to introduce visual expression, which is endowed with the target-informative visual guidance ability, as a source of target information. Nevertheless, this explanation has not fully satisfied the reviewers.
Concerning (b), the authors have furnished more ablation studies, which are expected to address the majority of the concerns.
With respect to (c), since the initial paper did not incorporate this aspect, during the rebuttal they have provided these comparisons. However, the improvements over the SoTAs are relatively slight.

---

### Decision · Program_Chairs · 2025-01-22

Reject